# PROFBENCH: MULTI-DOMAIN RUBRICS REQUIRING PROFESSIONAL KNOWLEDGE TO ANSWER AND JUDGE

**Zhilin Wang, Jaehun Jung, Ximing Lu, Shizhe Diao,**
**Ellie Evans, Jiaqi Zeng, Pavlo Molchanov, Yejin Choi, Jan Kautz, Yi Dong**
NVIDIA
{zhilinw, yidong}@nvidia.com

## ABSTRACT

Evaluating progress in large language models (LLMs) is often constrained by the challenge of verifying responses, limiting assessments to tasks like mathematics, programming, and short-form question-answering. However, many real-world applications require evaluating LLMs in processing professional documents, synthesizing information, and generating comprehensive reports in response to user queries. We introduce ProfBench: a set of over 7000 response-criterion pairs as evaluated by human-experts with professional knowledge across Physics PhD, Chemistry PhD, Finance MBA and Consulting MBA. We build robust and affordable LLM-Judges to evaluate ProfBench rubrics, by mitigating self-enhancement bias and reducing the cost of evaluation by 2-3 orders of magnitude, to make it fair and accessible to the broader community. Our findings reveal that ProfBench poses significant challenges even for state-of-the-art LLMs, with top-performing models like GPT-5-high achieving only 65.9% overall performance. Furthermore, we identify notable performance disparities between proprietary and open-weight models and provide insights into the role that extended thinking plays in addressing complex, professional-domain tasks.

🤗 **Data:** https://huggingface.co/datasets/nvidia/ProfBench
⚙ **Code:** https://github.com/NVlabs/ProfBench
🏆 **Leaderboard:** https://hf.co/spaces/nvidia/ProfBench

Table 1: Comparison of ProfBench with select rubric-based benchmarks. Rationales for the classification into good (✓) and poor (✗) on all aspects are discussed in the Introduction.

| Dataset | Diverse Domains | Professional Knowledge | Human Written Rubrics | Fair Grading |
|---|:---:|:---:|:---:|:---:|
| PaperBench (Starace et al., 2025) | ✗ | ✓ | ✓ | ✓ |
| HealthBench (Arora et al., 2025) | ✗ | ✓ | ✓ | ✓ |
| DeepResearch-Bench RACE (Du et al., 2025) | ✓ | ✗ | ✗ | ✗ |
| ProfBench (Ours) | ✓ | ✓ | ✓ | ✓ |

## 1 INTRODUCTION

For many problems, verifying the correctness of a solution can be much simpler than coming up with the solution. Take for instance, Sudoku - verifying a solution is trivial for anyone with a basic understanding of the rules (no repeated number in a row, column or small square) but the hardest initial positions can take experts a long time to solve. This insight inspired many to use Reinforcement Learning with Verified Rewards (RLVR; Lambert et al. 2025; DeepSeek-AI et al. 2025) by verifying LLM responses to tasks of such nature. However, there are only a limited set of tasks for which verified rewards can be easily constructed. One common domain is competition math, which have unique correct answers such as AIME 25 (White et al., 2025). Other common uses lay in competitive programming problems where generated code has to pass existing unit tests such as LiveCodeBench (Jain et al., 2024) or writing with precise instruction following (*e.g.,* checking if a particular word appears 5 times using a python script) in IFBench (Pyatkin et al., 2025). In other domains such as scientific question answers, RLVR is restricted to settings with a unique correct answer (Multiple-Choice Question or Short Answer Span), seen in popular evaluations like MMLU-Pro (Wang et al., 2024), GPQA (Rein et al., 2023) and HLE (Phan et al., 2025).

---

**ProfBench Finance MBA Example**

You are helping me assess the potential for a new business unit within a major investment bank. This unit is entirely dedicated to innovative finance for healthcare, as well as social impact and environmental challenges at the global level. One example we are studying to assess this opportunity is how the Global Alliance for Vaccination and Immunization (GAVI) has been able to raise money on capital markets through the International Finance Facility for Immunization (IFFIm).

I would like you to help answer the following questions:
- How was IFFIm able to raise money on capital markets for vaccination and immunization campaigns?
- How did IFFIm apply securitization?
- What were the factors that made it possible for IFFIm to raise money on capital markets?
- What risks and challenges does IFFIm have to address and overcome?
- How would you assess whether IFFIm has been effective and successful at raising funding for GAVI, and overall, has it been a success for their operations and health goals?
- Can IFFIm be viewed as a blueprint for investing in and funding other areas of global health, or other major social or environmental challenges? If yes, identify a short pipeline of 3 to 5 organizations/topics that could take a similar approach and use innovative finance to raise funding to advance their goals. Provide a robust context, starting with defining what Gavi and IFFIm are, how they work, and how they are both related. Discuss, in detail, the technical aspects of how IFFIm works, and describe what makes it possible and effective. The technical discussion should describe how it works, and provide a detailed overview of how it raised money until now (this should include a summary of past issuances, showing on what markets it raised and what investors subscribed). Also, discuss what elements made IFFIm possible, as well as what risks and challenges IFFIm faces. This should help you form a robust and documented view on what makes IFFIm a success (or not), and how it could act as a blueprint (or not) for other initiatives.

Regarding format, I am looking for the style of a detailed investment memo discussion. You can use tables and bullets, but the memo should be mostly text, and you should walk me through your reasoning. I don't want the output to be mostly tables and bullets.

*Extraction Rubric Sample*: States that a breach of IFFIm's liquidity policy could negatively impact IFFIm's rating profile.
*Reasoning Rubric Sample*: States that vaccines are one of the most successful and cost-effective health investments in the world.
*Style Rubric Sample (another task)*: Present findings clearly to allow for effective use.

**Deep Research-Bench Finance & Business Example**

What are the investment philosophies of Duan Yongping, Warren Buffett, and Charlie Munger?

Figure 1: Example from ProfBench (Finance MBA) is substantially more challenging and detailed compared with Deep Research-Bench Finance & Business (Du et al., 2025). More examples in §A.

The ease of verification should not limit the type of tasks we can train models to successfully complete. While some of these tests might predict how well PhD candidates (or other early-stage domain-experts) understand their field, people do not graduate from PhD programs (or become true-experts) simply based on how well they can pass these "exam-style" questions. Instead, they do so based on the merits of their original contributions in ways that are valuable to others - for instance, coming up with a technique to synthesize cancer treatment medication. To expand the set of tasks which can be evaluated with RLVR, there is a pressing need to cover challenging tasks with real-world value into a format can be verified. Grading problems using rubrics represent a promising path forward (Gunjal et al., 2025), with the central idea being to decompose a complex problem into several criteria that a good response needs to fulfill. A trivial example is that when asking an LLM for food recommendation for a Tues dinner with a vegan friend in New York City, there can be many good answers but they must fulfill a. open on Tues night; b. have vegan options; c. in New York City.

Recent works (Starace et al., 2025; Arora et al., 2025) show that humans are often tasked with coming up with these criteria while LLM-judges are used to determine if a response fulfills the criteria (given both cost and efficacy considerations). Specifically, Starace et al. (2025) demonstrate that this can be done for specialized domains requiring professional knowledge - *health* in evaluating patient-physician conversations while Arora et al. (2025) shows its usefulness for *machine learning* in reproducing ICML papers. However, there currently is no robust publicly available benchmark that makes use of rubric fulfillment across *diverse* professional domains beyond health and machine learning. DeepResearch-Bench RACE (Du et al., 2025) claims to curate PhD-level tasks across multiple domains, but many examples such as *What are the investment philosophies of Duan Yongping, Warren Buffett, and Charlie Munger?* or *How did Netflix manage to successfully adapt One Hundred Years of Solitude, a notoriously difficult book to bring to the screen?* can be answered by an educated generalist (i.e., college graduate or equivalent experience) with a few straightforward internet searches. It also falls short due to synthetically generated criteria that were not verified by expert humans— meaning that it's unclear if such criteria are *robust* anchors for scoring. For instance, there is a pervasive bias towards Gemini-2.5-Pro since the criteria and the reference 'high-quality' answer are both generated by this model. This results in Gemini-2.5-Pro being rated as by far the best performing—reaching more than 48.5 out of a maximum of 50 on each of the 4 axes, meaning >97%.

To support the community with robust rubrics for real-world, professional-level tasks, we propose ProfBench—a rubric-guided benchmark for professional-grade LLMs, curated by expert human professionals. ProfBench covers tasks across four domains—Chemistry, Physics, Consulting, Finance— annotated by human experts with either PhD degree (Chemistry and Physics), MBA (Consulting and Finance) or equivalent experience and are incumbent professionals in their respective domains. We summarize our main contributions below, which can also serve to inspire future directions in §H:

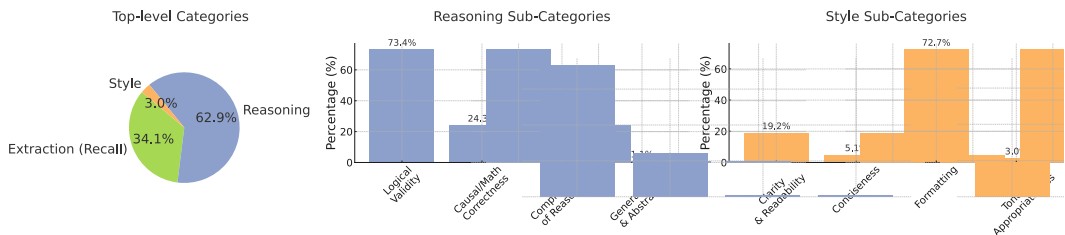

Figure 2: Distribution of rubrics by category. Reasoning dominates (62.9%), with most on logical validity and correctness. Extraction accounts for 34.1%, emphasizing accurate retrieval of information (w/o meaningful subcategories). Style is minor (3.0%), with a focus on formatting and clarity.

1. We introduce ProfBench, the first benchmark with expert-created rubrics across multiple professional domains including over 7000 response-criterion pairs involving Scientific Research and Business tasks. This benchmark is challenging - requiring PhD/MBA-level knowledge - with the strongest GPT-5-High model reaching only 65.9% performance.

2. We measure the performance of over 40 models in terms of their ability to generate high-quality responses to these tasks as well as to evaluate whether such responses fulfill various rubrics that experts determine a good response require as LLM-Judges. We analyze various trends across open/closed-source models, reasoning/instruct models and model size.

3. We propose methods to reduce the bias of LLM-Judges in favoring responses from specific providers as well as the cost of running the benchmark, in order to improve its accessibility. Our benchmark can be run with no more than 1% bias across 3 models from different providers and $12 (using o3 model), which is 2-3 orders of magnitude cheaper than existing rubric-based evaluation.

## 2 PROFBENCH OVERVIEW

ProfBench contains 7,347 human-written response-criterion pairs across 80 distinct tasks equally divided among 4 domains (Chemistry PhD, Physics PhD, Finance MBA and Consulting MBA). This is comparable to the 8,316 response-criterion pairs used in PaperBench (Starace et al., 2025) across 20 tasks as well as HealthBench (Arora et al., 2025) with 8,053 response-criterion pairs across hundreds of tasks. DeepResearch-Bench (Du et al., 2025) has 100 tasks (each with one reference response, and no human-written rubric) across 22 domains, equally split between Chinese and English, meaning there is only an average of 2.3 English tasks per domain. Rubric-based evaluations requiring annotation by human-experts tend to be small as each task is time-consuming and recruiting for such professionals (e.g., Physics PhD holders with industry experience) is difficult. We show an example in Fig. 1, a breakdown of rubrics by category in Fig. 2, and further descriptive statistics in §D.

## 3 DATA COLLECTION

**Annotator Recruitment** Annotators with PhD, MBA or equivalent work experience are recruited and managed by our vendor. Prior to their inclusion into the project, we check that they pass a test on their domain expertise and their understanding of the annotation task (e.g. performing under ambiguity and attention to detail). Each annotator is expected to work through an entire task - including Prompt Ideation, Rubric Creation and Response Annotation - as annotators are asked to create tasks that capture their personal expertise. Annotators are supported by one or more reviewers (with substantial task creation expertise in their domain) to review (several back-and-forth rounds possible if needed) and approve at each stage, before moving to the next. Each annotator spends around 10 to 20 hours on each task. To ensure diversity of prompts, each annotator is allowed to contribute no more than 5 tasks. We disallow the use of LLMs at any stage of the annotation process. Across the 80 tasks, 38 annotators from 8 countries were involved. 44.7 % of annotators hold a PhD, 18.4% hold MBA and others hold related degrees such as Bachelor of Science in Finance, Bachelor of Commerce or Bachelor of Business Administration along with work experience. Annotators have an average of 5.24 years of experience following graduation of their highest degree. Annotators are paid well-above minimum-wage following local standards, with hourly pay often exceeding full-time employment hourly pay in these professional fields to attract the most qualified annotators. Further details on annotator recruitment in §C.

**Prompt Curation** Our goal was to curate tasks that would be difficult even for the frontier LLMs at time of our collection in July 2025 (e.g. OpenAI o3; xAI Grok4; DeepSeek R1-0528). More specifically, we asked for tasks that annotators might ask their (junior) colleagues to help on, resulting in a multi-page report. Therefore, these questions often contain multiple related sub-questions as seen in the Finance MBA example in Fig 1. All prompts (and expected generations) must be written in English and only involve the text modality. We allow the annotator to use Internet search to identify documents that would support finding evidence for answers but disallow the use of proprietary documents (i.e. not findable on public internet). To ensure that prompts are created purposefully, we ask annotators to provide a brief rationale for how they come up with each prompt.

**Rubric Creation** For each task, we ask annotators to create around 15 to 60 criteria used to score the response of models. Each criteria should be independently usable to grade responses and collectively capture all aspects of the response quality. Each criterion contains a description of the criterion as well as a justification of the criterion (to encourage thoughtful creation). In addition, annotators are asked to assign an importance and one or more criteria types. The reviewer then provides overall feedback on the set of criteria as well as recommendation to keep or improve each criterion (details in §I). Among annotator-written criteria, 41.4% are marked as needing improvement at various stages, indicating the high quality standard that we set for when reviewing this data.

**Response Annotation** Our vendor generates responses with 3 models: OpenAI o3, Grok4 and DeepSeek R1-0528. These models represent the models that we believe to perform best on the tasks we collected at the start of annotations (July 2025), including both proprietary and open-weight models. The annotator scores each of the three responses on each criteria with either Yes or No, alongside a brief justification. Further details in §E.

## 4 BENCHMARKING MODELS AS LLM-JUDGES

**Task Formulation** For each criterion, we provide the LLM-Judge the response, the criterion and ask whether it fulfills the criterion in a binary fashion. In this sense, the task is formulated similar to a two-class Natural Language Inference/Recognizing Textual Entailment task (Bowman et al., 2015) with two possible answers (Entailment or Contradiction). Following PaperBench (Starace et al., 2025), we do not provide the original task prompt to the LLM-judge, as the criteria are designed to be used independently of the prompt, and further providing the prompt might confuse the LLM-Judge. Prompt templates are in §B.

### 4.1 EVALUATION

**Agreement with Human Annotations** To evaluate LLM-Judges, we use Macro-F1 based on the ground-truth human-labeled criterion-fulfillment and the model-predicted criterion-fulfillment (both binary) as used by HealthBench (Arora et al., 2025) and PaperBench (Starace et al., 2025).

**Bias** However, another important aspect to consider is fairness to various model responses since LLMs are known to have self-enhancement bias (Zheng et al., 2023), meaning that they allocate higher scores to their own responses or responses from the same model family. We formulate a bias-index by first calculating the bias for each model using $\frac{1}{N}\sum_{i=1}^{N} c_i^{model} - c_i^{human}$ where $N$ is total number of criteria and $c_i$ is the criterion-fulfillment predicted by model or labeled by human. Then, we calculate the min and max bias across three models (o3, Grok4 and R1-0528) before calculating the bias-index by taking the difference between max-bias minus the min-bias. A low bias-index means that an LLM-Judge does not overly-reward/penalize any model's responses, relative to human annotated ground-truths and other models. Overall performance is Macro-F1 minus Bias-Index.

**Cost** We calculate the upper-bound cost of running the full LLM-Judge evaluation, using the number of input and output tokens multiplied by their public cost (OpenRouter, 2025), excluding discounts due to caching and other methods. In practice, caching alone will reduce the cost to as low as one-tenth as stated cost for judges whose costs are dominated by input tokens. Cost is useful when comparing two judges with similar performance, as the cheaper judge can be accessible for more.

**Inference Setup**  Following HealthBench (Arora et al., 2025), we use GPT-4.1 as a judge of response-criterion fulfillment in early experiments to identify an optimal prompt template used across all models subsequently among possible templates (See §B). Our exploration supports generating only 1 completion token (Yes or No) for non-reasoning LLMs and up to 32,000 tokens for reasoning LLMs (and expect the post-thinking-trace response to be either Yes or No). This formulation means that non-reasoning LLMs will be substantially cheaper and faster than reasoning LLMs (approximately two to three orders of magnitude in practice). Following best practices (Yang et al., 2025; DeepSeek-AI et al., 2025), we set temperature to 0.6 / top-p 0.95 for reasoning LLMs (when it can be set) and temperature 0 / top-p 0 (i.e. greedy decoding) for non-reasoning LLMs. Our experiments with GPT-4.1 Judge also suggests that both Macro-F1 and Bias-Index are highly consistent, differing no more than 0.2% across three independent runs - therefore we only run with each judge once to save cost. Note that for judges, we do not require web search or file upload as human annotators were explicitly asked to create criteria that can be independently used to grade responses. For experimental purpose, we only use half of the dataset as we plan for this to be the public dataset, while we keep the remaining half as the private dataset in order to mitigate test contamination (Han et al., 2025).

**Human Validation**  We carry out a validation experiment on a portion of the dataset (1127 response-criterion pairs). Specifically, we ask two other annotators with the same expertise (e.g. Chemistry PhD) to re-annotate the response-criterion fulfillment. Based on such annotations, we find inter-annotator agreement to be Fleiss' $\kappa = 0.912$, suggesting excellent agreement among annotators.

## 4.2  RESULTS

**Are Closed Source Models better than Open Weight ones?**  Across the board, the best LLM-Judge models in Tab. 2 are proprietary models. GPT-4.1 is the best model that outputs only 1 token per task at 75.4%, while Gemini-2.5-Pro takes the crown for Reasoning LLM Judges at 78.2% (with its Flash sibling only 0.1% behind). However, open-weights are often not far behind. In the non-thinking category, Kimi-K2-0711 is only 0.2% behind while GPT-OSS-120B-low is 1.5% behind within the reasoning category. Such a small gap is especially impressive given the cost difference. Kimi-K2 costs a mere 7.16% of GPT-4.1 while GPT-OSS-120B-low is only 1.21% of Gemini-2.5-Pro ($0.50 vs $41.46). This means that using open-weight models can allow experiments with evaluating model responses to be much cheaper and hence accessible to many more. Such accessibility allows more researchers from a wider background to iterate with methods to train rubric-following behavior.

**Does Model Size Matter?**  Within the same model family, larger models generally perform better than smaller models - but the improvement plateaus after a certain size. For instance, the improvement between GPT-4.1-nano and GPT-4.1-mini was large at +20.8% while the further improvement to GPT-4.1 was much smaller at +0.5%. A similar trend can be observed with the Llama-3.1/3.2 series going from 1B to 405B. However, the jump from llama-3.1-70B to 3.3-70B (+3.4%) was much larger than the jump from llama-3.1-70B to llama-3.1-405B (+0.9%), indicating the importance of improvements in post-training recipe beyond model size alone. This is also supported by the minuscule 0.1% model performance gains between Gemini-2.5-Flash-Lite and Gemini-2.5-Flash (non-thinking) as well as Gemini-2.5-Flash (thinking) and Gemini-2.5-Pro (thinking).

**To Think or Not to Think?**  When comparing the same model (e.g. Gemini-2.5-Flash, Claude-Sonnet-4, DeepSeek-V3.1) that can have thinking enabled or disabled, enabling thinking typically leads to some improvements, although the magnitude of the gain varies: 0.7% for Claude-Sonnet-4 to 4.8% for Gemini-2.5-Flash. This is further supported by the OpenAI GPT-5 series, for which going from minimal reasoning to low reasoning substantially increases performance by 4.4% to 18.6%, with a larger gain for the smallest GPT-5-nano. In contrast, the gain between low, medium and high reasoning effort is not consistent, with o4-mini, GPT-5-nano and gpt-oss-20b/120b showing the best performance at low reasoning effort. A possible explanation is that increasing reasoning effort generally increases alignment with human annotations but also increases bias to specific models - as observed consistently across all models (except for o3). This could be a result of greater self-enhancement bias with more thinking, as the bias towards OpenAI o3 responses generally increases. Furthermore, thinking with greater effort seems to lead to largest improvement in Physics, Chemistry and criteria related to Style (e.g. Rounds all stock prices and equity values to two decimal places). These criteria could be harder to judge without extending thinking, since they often entail multi-step reasoning to ensure that the response fulfills the criteria, instead of a simple answer-matching.

Table 2: Evaluation of LLM-Judges. Higher is better for Macro-F1 and Overall while closer to 0 is better for Bias Index. Sensitivity and Qualitative analyses of LLM-Judges are in §J.

| Model | Agreement with Human Annotations (Macro-F1) ↑ | | | | | | | | Bias Index ↓ | | | | Overall ↑ | Tokens ↓ | | |
|---|---|---|---|---|---|---|---|---|---|---|---|---|---|---|---|---|
| | Phys | Chem | Fin | Consult | Extract | Reason | Style | All | o3 | R1-05 | Grok4 | Max-Min | MF1 - BI | In | Out | $ |
| *Instruct LLM Judge w. 1 Output token per task* | | | | | | | | | | | | | | | | |
| OpenAI/GPT-4.1 | 80.9 | 69.2 | 71.0 | 80.0 | 79.8 | 74.4 | 65.8 | 76.3 | 5.5 | 4.6 | 5.0 | 0.9 | 75.4 | 1619 | 1 | 11.31 |
| OpenAI/GPT-4.1-mini | 83.9 | 67.3 | 69.1 | 80.6 | 79.2 | 74.7 | 69.8 | 76.4 | -0.2 | 1.2 | -0.3 | 1.5 | 74.9 | 1619 | 1 | 2.26 |
| OpenAI/GPT-4.1-nano | 69.8 | 62.9 | 66.7 | 68.4 | 71.0 | 65.6 | 63.5 | 67.9 | -14.5 | -2.1 | -0.7 | 13.8 | 54.1 | 1619 | 1 | 0.56 |
| Google/Gemini-2.5-Flash | 82.9 | 67.3 | 70.8 | 79.6 | 79.2 | 74.5 | 67.7 | 76.3 | -4.2 | -6.6 | -7.1 | 2.9 | 73.4 | 1779 | 1 | 1.87 |
| Google/Gemini-2.5-Flash-Lite | 83.6 | 68.2 | 68.2 | 80.6 | 77.9 | 75.0 | 71.0 | 76.4 | -1.1 | 2.0 | 0.6 | 3.1 | 73.3 | 1779 | 1 | 0.62 |
| Anthropic/claude-sonnet-4 | 85.0 | 66.9 | 68.1 | 76.3 | 77.6 | 73.3 | 64.1 | 75.2 | -6.5 | -5.2 | -10.2 | 5.0 | 70.2 | 1913 | 1 | 20.06 |
| anthropic/claude-3.5-haiku | 78.9 | 67.2 | 71.2 | 76.7 | 76.9 | 73.3 | 65.4 | 74.9 | -1.7 | 0.7 | -1.4 | 2.4 | 72.5 | 1913 | 1 | 5.35 |
| Open-weight | | | | | | | | | | | | | | | | |
| Qwen/Qwen3-235B-A22B-Instruct-2507 | 86.5 | 69.3 | 69.3 | 79.6 | 79.2 | 76.0 | 64.6 | 77.3 | 3.8 | 2.2 | 1.6 | 2.2 | 75.1 | 1779 | 1 | 0.48 |
| Qwen/Qwen3-30B-A3B-instruct-2507 | 82.0 | 68.3 | 67.3 | 79.7 | 76.5 | 74.5 | 64.7 | 75.5 | 4.7 | 7.1 | 5.3 | 2.4 | 73.1 | 1778 | 1 | 0.32 |
| MoonshotAI/Kimi-K2-Instruct-0905 | 84.5 | 69.9 | 67.5 | 81.9 | 80.2 | 75.5 | 65.9 | 77.0 | 7.5 | 6.1 | 5.2 | 2.3 | 74.7 | 1623 | 1 | 0.81 |
| MoonshotAI/Kimi-K2-Instruct-0711 | 85.3 | 69.5 | 68.3 | 82.3 | 80.3 | 76.1 | 66.4 | 77.6 | 7.1 | 6.1 | 4.7 | 2.4 | 75.2 | 1636 | 1 | 0.81 |
| DeepSeek-AI/DeepSeek-V3.1 | 79.6 | 68.2 | 68.3 | 78.7 | 77.4 | 73.9 | 65.8 | 75.2 | 0.2 | -1.5 | -2.2 | 2.4 | 72.8 | 1586 | 1 | 1.11 |
| DeepSeek-AI/DeepSeek-V3-0324 | 84.5 | 68.0 | 67.0 | 78.3 | 77.7 | 76.4 | 63.5 | 75.7 | 1.5 | 2.4 | -0.7 | 3.1 | 72.6 | 1585 | 1 | 1.11 |
| nvidia/llama-3.1-nemotron-nano-8b-v1 | 56.5 | 59.5 | 57.3 | 56.7 | 61.3 | 58.6 | 59.1 | 59.3 | -28.5 | -26.5 | -30.0 | 3.5 | 55.8 | 1633 | 1 | 0.09 |
| nvidia/llama-3.3-nemotron-super-49b-v1 | 77.2 | 65.1 | 70.2 | 72.1 | 74.9 | 75.0 | 64.1 | 72.3 | -15.7 | -12.2 | -13.0 | 3.5 | 68.8 | 1637 | 1 | 0.74 |
| nvidia/llama-3.1-nemotron-ultra-253b-v1 | 84.8 | 63.6 | 66.6 | 61.8 | 72.6 | 67.8 | 57.8 | 69.6 | -10.0 | -11.4 | -9.2 | 2.2 | 67.4 | 1637 | 1 | 3.43 |
| meta/llama-4-maverick-17b-128e-instruct | 64.9 | 66.7 | 73.4 | 76.4 | 76.5 | 70.4 | 67.9 | 72.4 | -14.3 | -10.5 | -9.8 | 4.5 | 67.9 | 1566 | 1 | 0.82 |
| meta/llama-4-scout-17b-16e-instruct | 60.4 | 69.4 | 71.3 | 75.6 | 76.2 | 69.9 | 62.0 | 71.8 | -14.5 | -10.2 | -8.6 | 5.9 | 65.9 | 1565 | 1 | 0.44 |
| meta/llama-3.1-405b-instruct | 85.1 | 69.1 | 67.6 | 81.7 | 77.7 | 75.5 | 65.5 | 77.0 | 11.5 | 6.1 | 9.4 | 5.4 | 71.6 | 1628 | 1 | 4.54 |
| meta/llama-3.3-70b-instruct | 84.6 | 66.5 | 71.6 | 79.1 | 78.1 | 75.4 | 64.6 | 76.7 | -3.1 | -0.8 | -3.4 | 2.6 | 74.1 | 1628 | 1 | 0.22 |
| meta/llama-3.1-70b-instruct | 82.1 | 66.7 | 72.6 | 76.0 | 77.5 | 73.9 | 64.7 | 75.4 | -6.2 | -1.5 | -4.1 | 4.7 | 70.7 | 1628 | 1 | 0.22 |
| meta/llama-3.1-8b-instruct | 76.2 | 69.3 | 70.2 | 71.0 | 76.6 | 71.5 | 61.7 | 73.2 | -4.0 | 6.1 | -1.5 | 10.1 | 63.1 | 1628 | 1 | 0.09 |
| meta/llama-3.2-3b-instruct | 67.6 | 63.8 | 59.7 | 66.1 | 68.8 | 64.6 | 54.6 | 66.2 | 8.8 | 16.7 | 13.1 | 7.9 | 58.3 | 1628 | 1 | 0.02 |
| meta/llama-3.1-1b-instruct | 31.9 | 48.4 | 44.9 | 55.8 | 47.8 | 43.2 | 46.2 | 45.7 | 31.0 | 33.1 | 37.2 | 6.2 | 39.5 | 1628 | 1 | 0.02 |
| *Reasoning LLM Judge* | | | | | | | | | | | | | | | | |
| OpenAI/GPT-5 | | | | | | | | | | | | | | | | |
| - high | 90.2 | 68.2 | 69.4 | 80.9 | 78.3 | 76.7 | 79.1 | 78.3 | 1.0 | -0.8 | -1.3 | 2.3 | 76.0 | 1618 | 668 | 30.34 |
| - med | 89.2 | 67.9 | 69.0 | 80.9 | 78.1 | 76.3 | 77.3 | 77.9 | 0.0 | -0.9 | -1.2 | 1.2 | 76.7 | 1619 | 287 | 17.06 |
| - low | 88.6 | 69.3 | 69.0 | 80.9 | 78.1 | 76.6 | 79.4 | 78.1 | 0.3 | -1.5 | -1.4 | 1.8 | 76.3 | 1618 | 130 | 11.58 |
| - minimal | 86.8 | 68.6 | 71.2 | 77.5 | 78.9 | 75.2 | 64.8 | 77.0 | -0.5 | -5.6 | -5.0 | 5.1 | 71.9 | 1618 | 7 | 7.29 |
| OpenAI/GPT-5-mini | | | | | | | | | | | | | | | | |
| - high | 84.5 | 69.2 | 70.4 | 82.8 | 78.4 | 75.9 | 74.1 | 77.7 | 6.6 | 4.2 | 4.6 | 2.4 | 75.3 | 1619 | 497 | 4.88 |
| - med | 83.3 | 68.2 | 69.9 | 81.5 | 78.1 | 74.6 | 72.8 | 76.7 | 6.3 | 4.0 | 4.3 | 2.3 | 74.4 | 1618 | 228 | 3.00 |
| - low | 82.9 | 68.5 | 70.3 | 81.7 | 77.4 | 74.6 | 78.0 | 76.8 | 5.9 | 3.8 | 4.6 | 2.1 | 74.7 | 1618 | 92 | 2.05 |
| - minimal | 81.7 | 64.0 | 69.1 | 76.0 | 75.9 | 72.5 | 58.8 | 73.8 | -4.0 | -6.2 | -11.1 | 7.1 | 66.7 | 1618 | 7 | 1.46 |
| OpenAI/GPT-5-nano | | | | | | | | | | | | | | | | |
| - high | 86.8 | 67.6 | 68.7 | 79.8 | 77.6 | 75.1 | 74.0 | 76.9 | 5.3 | 0.3 | 3.1 | 5.0 | 71.9 | 1618 | 1309 | 2.11 |
| - med | 85.6 | 67.0 | 68.7 | 79.7 | 77.1 | 74.3 | 78.3 | 76.4 | 3.4 | -0.3 | 1.7 | 3.7 | 72.7 | 1618 | 479 | 0.95 |
| - low | 83.5 | 67.6 | 68.6 | 77.7 | 76.9 | 73.5 | 70.9 | 75.4 | 2.4 | 0.6 | 1.9 | 1.8 | 73.6 | 1619 | 141 | 0.48 |
| - minimal | 68.8 | 55.3 | 60.9 | 63.0 | 65.8 | 62.1 | 54.3 | 63.2 | -18.7 | -19.6 | -26.9 | 8.2 | 55.0 | 1618 | 7 | 0.29 |
| OpenAI/o3 | | | | | | | | | | | | | | | | |
| - high | 88.3 | 68.2 | 69.3 | 81.1 | 79.1 | 76.1 | 75.3 | 77.9 | 2.0 | 0.5 | 0.8 | 1.5 | 76.4 | 1618 | 350 | 21.04 |
| - med | 89.3 | 68.1 | 68.9 | 81.0 | 79.3 | 76.4 | 76.9 | 78.2 | 3.0 | 0.8 | 1.5 | 2.2 | 76.0 | 1618 | 207 | 17.05 |
| - low | 88.9 | 69.3 | 70.3 | 81.9 | 79.7 | 76.8 | 76.7 | 78.7 | 3.8 | 1.5 | 2.6 | 2.3 | 76.4 | 1618 | 98 | 14.01 |
| OpenAI/o4-mini | | | | | | | | | | | | | | | | |
| - high | 88.5 | 68.9 | 70.5 | 81.5 | 78.7 | 76.8 | 76.5 | 78.4 | 4.5 | 2.7 | 1.9 | 2.6 | 75.8 | 1618 | 308 | 10.93 |
| - med | 88.1 | 69.6 | 70.8 | 81.6 | 78.9 | 76.8 | 74.1 | 78.6 | 4.0 | 2.8 | 1.2 | 2.8 | 75.8 | 1618 | 228 | 9.70 |
| - low | 88.6 | 70.1 | 70.1 | 81.0 | 78.8 | 76.8 | 74.1 | 78.5 | 3.4 | 3.3 | 1.7 | 1.7 | 76.8 | 1618 | 104 | 7.80 |
| xAI/grok-4 | 86.1 | 68.5 | 70.7 | 80.8 | 78.5 | 76.3 | 75.2 | 77.7 | 0.7 | 2.5 | 1.8 | 1.8 | 75.9 | 1549 | 812 | 58.7 |
| xAI/grok-3-mini | 85.8 | 66.9 | 69.4 | 82.0 | 78.1 | 75.3 | 75.2 | 77.2 | 4.5 | 2.4 | 2.9 | 2.1 | 75.1 | 1549 | 633 | 2.72 |
| Anthropic/claude-sonnet-4-20250514 | 75.7 | 66.3 | 69.9 | 77.8 | 77.5 | 73.3 | 66.0 | 74.0 | -11.2 | -8.1 | -10.7 | 3.1 | 70.9 | 1940 | 810 | 62.63 |
| Google/Gemini-2.5-Pro | 87.3 | 70.2 | 71.9 | 82.6 | 81.3 | 77.4 | 76.8 | 79.2 | 3.1 | 2.8 | 2.1 | 1.0 | 78.2 | 1779 | 967 | 41.46 |
| Google/Gemini-2.5-Flash (Thinking) | 87.0 | 68.7 | 71.6 | 81.2 | 80.1 | 76.7 | 74.6 | 78.4 | 2.3 | 2.5 | 2.2 | 0.3 | 78.1 | 1779 | 695 | 7.92 |
| Google/Gemini-2.5-Flash-Lite (Thinking) | 83.7 | 67.0 | 72.2 | 81.9 | 78.7 | 75.9 | 79.1 | 77.5 | -1.1 | 0.2 | -2.6 | 2.8 | 74.7 | 1779 | 1670 | 2.95 |
| Open-weight | | | | | | | | | | | | | | | | |
| OpenAI/gpt-oss-20b | | | | | | | | | | | | | | | | |
| - high | 89.3 | 68.7 | 68.5 | 80.7 | 77.8 | 76.5 | 77.7 | 77.9 | 3.3 | -0.2 | 0.9 | 3.5 | 74.4 | 1679 | 465 | 0.46 |
| - medium | 87.7 | 68.3 | 69.7 | 80.9 | 78.5 | 76.3 | 76.2 | 77.8 | 3.6 | 1.1 | 0.6 | 3.0 | 74.8 | 1683 | 216 | 0.35 |
| - low | 85.4 | 69.3 | 70.8 | 79.2 | 77.6 | 76.3 | 71.1 | 77.5 | 0.4 | -0.3 | 1.6 | 1.9 | 75.6 | 1677 | 85 | 0.28 |
| OpenAI/gpt-oss-120b | | | | | | | | | | | | | | | | |
| - high | 89.5 | 68.9 | 69.7 | 80.8 | 78.9 | 76.7 | 80.8 | 78.4 | 1.6 | -1.4 | 0.3 | 3.0 | 75.4 | 1683 | 439 | 0.88 |
| - med | 88.1 | 67.4 | 70.5 | 79.9 | 79.6 | 76.0 | 75.3 | 77.7 | 0.6 | -1.3 | -0.9 | 1.9 | 75.8 | 1683 | 196 | 0.63 |
| - low | 86.0 | 67.2 | 72.1 | 79.0 | 79.2 | 75.7 | 72.4 | 77.3 | -1.0 | -1.6 | -1.5 | 0.6 | 76.7 | 1683 | 84 | 0.50 |
| - high for physics/chemistry/style + low for others | 89.5 | 68.9 | 72.2 | 79.7 | 79.2 | 76.9 | 80.8 | 78.7 | -0.5 | -0.9 | -1.0 | 0.5 | 78.2 | 1683 | 282 | 0.70 |
| DeepSeek-AI/DeepSeek-V3.1 (Thinking) | 84.3 | 69.3 | 70.8 | 80.3 | 78.9 | 75.6 | 72.0 | 77.3 | 3.2 | 3.3 | 2.6 | 0.7 | 76.6 | 1587 | 657 | 2.94 |
| DeepSeek-AI/DeepSeek-R1-0528 | 79.6 | 65.1 | 68.5 | 71.6 | 74.7 | 70.9 | 64.1 | 72.2 | -11.6 | -9.3 | -8.8 | 2.8 | 69.4 | 1601 | 693 | 3.05 |
| Qwen/Qwen3-30B-A3B-Thinking-2507 | 46.7 | 35.9 | 45.4 | 35.8 | 42.1 | 41.2 | 35.3 | 41.5 | -0.2 | -1.3 | 0.4 | 1.7 | 39.8 | 1780 | 742 | 1.10 |
| Qwen/Qwen3-235B-A22B-Thinking-2507 | 87.2 | 67.9 | 69.0 | 80.4 | 79.3 | 75.6 | 74.3 | 77.3 | -1.0 | -1.8 | -1.5 | 0.8 | 76.5 | 1782 | 1245 | 1.84 |

**Best Performing Judge Overall**  In selecting the optimal judge for evaluating report-generation models, we consider both the overall score (which consider alignment with humans and freedom from bias) and the cost of running the LLM Judge, which influences the accessibility of the benchmark. Therefore, we decided to use GPT-OSS-120B as the judge since it does well on both. We also noticed that the high reasoning effort version does better on Physics, Chemistry and Style-related criteria, while the low reasoning effort version does better on others. Therefore, we decided to alter the reasoning effort based on the domain/criterion type encountered, inspired by Jung et al. (2025) to balance quality and cost. The resulting judge performs as well as the best proprietary model (Gemini-2.5-Pro) at 78.2% Overall while costing only 1.68% of Gemini-2.5-Pro's cost. This means ProfBench-Judge only costs $0.70 vs. $1320 for PaperBench JudgeEval (Starace et al., 2025).

## 5 BENCHMARKING MODELS AS REPORT-GENERATORS

**Task Formulation**  We formalize the task as given a prompt alongside grounding documents, generate a response that addresses the task prompt. The use of grounding documents is inspired by how human professionals commonly work when tackling real-world tasks, which is a formulation

that has not been applied in popular benchmarks such as HLE, GPQA and MMLU-Pro. Based on the generated report, we use the best-performing GPT-OSS-120B Judge (high reasoning effort for Physics/Chemistry/Style-related criteria and low for all others) from §4.2 to grade the response on each criterion. Inspired by HealthBench (Arora et al., 2025) and PaperBench (Starace et al., 2025), we calculate the criteria-fulfillment rate of each response, weighted by their criterion importance (1 for additional, 2 for minor, 3 for major and 4 for critical). To validate this scoring schema, we calculate the predicted performance of the three models with human-annotated responses (o3, Grok4 and R1-0528). We find that this schema is capable of scoring each model with only a 0.7 to 1.3% gap between judge-predicted and human-annotated performance across 3 models. Further details in §E.

**Inference Setup**   Our inference setup largely follows that of §4.1. As we expect long generations, we generate up to 32,000 tokens for non-reasoning LLMs (max possible sequence length for GPT-4.1) and 64,000 for reasoning LLMs (max possible sequence length for Claude 4 Sonnet and Gemini 2.5 Pro). To estimate the performance variance, we repeat the generation 16 times and report the average alongside the standard deviation, following Bercovich et al. (2025). Further details in §E.

## 5.1   RESULTS

Table 3: Evaluation of LLMs as report-generators. Higher is better for Score and Accuracy.

| | | | Score | | | Accuracy by Criterion Type | | | Length | Tokens | | |
|---|---|---|---|---|---|---|---|---|---|---|---|---|
| Model | Physics | Chemistry | Finance | Consulting | Overall | Extract/Recall | Reasoning | Style | Chars. | In | Out | $ |
| *Reasoning LLM* | | | | | | | | | | | | |
| Closed-Source Reasoning | | | | | | | | | | | | |
| OpenAI/GPT-5 (high) | 49.3 (2.6) | 70.6 (2.1) | 63.7 (2.8) | 80.0 (2.0) | **65.9** (1.1) | 64.4 (2.3) | 66.2 (1.1) | 65.3 (7.7) | 5451 | 23758 | 14583 | 112.34 |
| OpenAI/GPT-5-mini (high) | 50.8 (2.8) | 63.6 (3.5) | 51.6 (2.8) | 75.4 (3.1) | 60.3 (1.2) | 56.7 (1.6) | 60.1 (1.0) | 68.2 (6.3) | 9018 | 26859 | 18038 | 27.39 |
| OpenAI/GPT-5-nano (high) | 42.2 (3.7) | 44.6 (2.8) | 44.6 (3.3) | 69.0 (2.0) | 50.1 (0.9) | 46.6 (1.9) | 48.3 (1.2) | 58.9 (6.5) | 9796 | 28549 | 25189 | 7.36 |
| OpenAI/o3 | 46.1 (2.8) | 61.8 (2.2) | 60.9 (2.8) | 76.8 (1.9) | 61.4 (1.3) | 60.4 (1.9) | 61.8 (1.3) | 63.0 (5.0) | 4158 | 18445 | 4709 | 47.72 |
| OpenAI/o4-mini | 45.5 (1.9) | 58.5 (2.2) | 54.7 (2.5) | 74.4 (2.7) | 58.2 (1.0) | 55.8 (1.7) | 58.3 (1.8) | 61.0 (5.6) | 3886 | 31679 | 4763 | 35.71 |
| Google/Gemini-2.5-Pro | 46.8 (2.2) | 66.3 (2.0) | 54.0 (3.2) | 74.2 (1.5) | 60.3 (1.3) | 61.4 (2.1) | 59.3 (1.4) | 66.8 (6.1) | 7449 | 6086 | 7950 | 55.75 |
| Google/Gemini-2.5-Flash (Thinking) | 45.0 (2.9) | 61.8 (3.5) | 53.5 (3.0) | 69.9 (2.3) | 57.6 (1.5) | 58.0 (2.8) | 57.6 (2.0) | 61.1 (6.8) | 12047 | 6086 | 12030 | 20.42 |
| Google/Gemini-2.5-Flash-Lite (Thinking) | 31.7 (2.1) | 53.1 (2.9) | 44.6 (3.8) | 68.0 (2.5) | 49.4 (1.3) | 48.3 (2.3) | 48.8 (1.4) | 54.0 (5.5) | 10058 | 6086 | 18584 | 5.15 |
| xAI/grok-4-0709 | 33.6 (2.4) | 62.2 (3.6) | 44.3 (3.1) | 73.4 (2.5) | 53.4 (1.5) | 51.9 (3.2) | 51.6 (1.7) | 64.1 (7.7) | 5380 | 13481 | 9885 | 122.78 |
| Anthropic/claude-sonnet-4 (Thinking) | 43.9 (2.3) | 57.1 (2.3) | 50.8 (2.6) | 71.4 (2.4) | 55.8 (0.9) | 53.8 (1.9) | 54.0 (1.3) | 61.8 (4.8) | 3866 | 51044 | 6916 | 164.39 |
| Open-weight Reasoning | | | | | | | | | | | | |
| OpenAI/gpt-oss-120b | 49.1 (2.4) | 55.3 (3.4) | 45.5 (1.7) | 69.4 (2.5) | 54.9 (1.4) | 48.7 (2.5) | 55.5 (1.4) | 59.0 (5.6) | 7442 | 11606 | 4572 | 1.35 |
| OpenAI/gpt-oss-20b | 41.4 (2.4) | 46.5 (3.5) | 39.8 (2.8) | 66.0 (2.4) | 48.4 (1.1) | 40.9 (1.4) | 48.2 (1.2) | 56.2 (7.6) | 5331 | 11600 | 4705 | 0.75 |
| DeepSeek-AI/DeepSeek-V3.1 (Thinking) | 44.8 (3.0) | 59.8 (3.3) | 43.3 (2.4) | 67.4 (2.1) | 53.8 (1.2) | 51.1 (2.4) | 53.0 (1.3) | 60.5 (6.7) | 5239 | 11258 | 7486 | 5.27 |
| Qwen/Qwen3-235B-A22B-Thinking-2507 | 45.1 (2.1) | 61.4 (2.6) | 42.3 (2.5) | 67.3 (2.4) | 54.0 (1.1) | 51.4 (1.9) | 51.6 (1.7) | 61.9 (6.6) | 6046 | 12442 | 9256 | 2.47 |
| Qwen/Qwen3-30B-A3B-Thinking-2507 | 34.4 (2.5) | 45.4 (2.4) | 36.8 (2.9) | 61.8 (2.2) | 44.6 (1.4) | 40.4 (1.5) | 42.3 (1.9) | 63.9 (4.8) | 4757 | 12339 | 9027 | 2.16 |
| *Instruct/Non-Reasoning LLM* | | | | | | | | | | | | |
| Closed-Source Instruct | | | | | | | | | | | | |
| OpenAI/GPT-4.1 | 44.7 (2.5) | 55.2 (2.8) | 54.0 (2.4) | 73.2 (2.2) | 56.8 (1.0) | 56.7 (1.5) | 56.7 (1.7) | 58.4 (8.8) | 6451 | 18427 | 2152 | 34.60 |
| OpenAI/GPT-4.1-mini | 45.1 (3.0) | 53.0 (3.0) | 49.1 (3.4) | 67.5 (2.4) | 53.7 (1.2) | 50.3 (1.8) | 53.2 (1.4) | 52.8 (7.2) | 6921 | 29469 | 2218 | 9.82 |
| OpenAI/GPT-4.1-nano | 24.8 (2.1) | 40.8 (3.3) | 33.4 (2.9) | 58.2 (3.0) | 39.3 (1.0) | 34.9 (2.0) | 38.4 (1.1) | 53.5 (5.9) | 6359 | 35561 | 1966 | 2.78 |
| Google/Gemini-2.5-Flash | 44.6 (3.0) | 59.4 (2.7) | 54.3 (2.9) | 68.8 (2.4) | 56.8 (1.3) | 57.1 (1.6) | 56.1 (1.4) | 53.2 (5.7) | 21612 | 6086 | 5936 | 10.67 |
| Google/Gemini-2.5-Flash-Lite | 29.8 (3.0) | 49.0 (2.3) | 44.0 (2.4) | 63.7 (2.3) | 46.6 (1.2) | 47.4 (2.3) | 45.0 (1.3) | 48.6 (6.7) | 24167 | 6086 | 7787 | 2.33 |
| Anthropic/claude-sonnet-4 | 40.7 (2.2) | 54.2 (3.0) | 49.5 (3.5) | 69.6 (2.0) | 53.5 (1.0) | 55.3 (2.2) | 51.1 (1.9) | 54.2 (6.8) | 4068 | 51016 | 1398 | 111.37 |
| Anthropic/claude-3.5-haiku | 12.0 (1.8) | 24.7 (2.6) | 27.7 (3.5) | 46.3 (3.0) | 27.6 (1.2) | 31.2 (2.3) | 24.7 (1.3) | 49.4 (6.7) | 1784 | 34475 | 576 | 19.13 |
| Open-weight Instruct | | | | | | | | | | | | |
| Qwen/Qwen3-235B-A22B-Instruct-2507 | 45.6 (2.4) | 55.8 (3.5) | 45.7 (2.6) | 69.6 (2.2) | 54.2 (1.2) | 51.0 (1.7) | 52.9 (1.6) | 66.2 (6.5) | 11400 | 12450 | 4244 | 1.47 |
| Qwen/Qwen3-30B-A3B-Instruct-2507 | 41.6 (1.8) | 47.9 (3.1) | 42.3 (2.5) | 65.5 (2.7) | 49.3 (0.7) | 44.5 (1.6) | 48.0 (1.3) | 59.1 (5.2) | 11167 | 12490 | 4021 | 0.95 |
| MoonshotAI/Kimi-K2-Instruct-0905 | 40.4 (2.5) | 50.2 (2.9) | 48.8 (2.7) | 65.9 (1.9) | 51.3 (1.1) | 51.2 (2.1) | 50.0 (1.4) | 63.4 (5.9) | 4817 | 11462 | 1562 | 3.36 |
| DeepSeek-AI/DeepSeek-V3.1 | 45.8 (2.0) | 55.9 (2.9) | 45.2 (3.0) | 67.1 (2.4) | 53.5 (1.4) | 50.8 (2.1) | 52.7 (1.8) | 59.1 (4.8) | 7792 | 11231 | 2407 | 2.67 |
| Meta/llama-4-maverick | 35.2 (2.1) | 35.8 (3.5) | 34.2 (2.5) | 52.5 (2.6) | 39.4 (1.4) | 39.3 (2.7) | 36.5 (1.5) | 46.2 (5.9) | 4223 | 14604 | 1191 | 1.86 |
| meta/llama-4-scout | 23.4 (1.8) | 34.6 (2.6) | 33.4 (1.7) | 50.3 (2.4) | 35.4 (1.2) | 35.1 (1.7) | 33.3 (1.4) | 42.3 (7.3) | 3612 | 16675 | 1039 | 1.05 |
| *Inference parameters* | | | | | | | | | | | | |
| OpenAI/GPT-5 | | | | | | | | | | | | |
| - high reasoning | 49.3 (2.6) | 70.6 (2.1) | 63.7 (2.8) | 80.0 (2.0) | 65.9 (1.1) | 64.4 (2.3) | 66.2 (1.1) | 65.3 (7.7) | 5451 | 23758 | 14583 | 112.34 |
| - medium reasoning (default) | 49.9 (2.0) | 69.0 (3.4) | 63.8 (2.7) | 78.0 (2.7) | 65.2 (1.8) | 63.7 (2.8) | 65.6 (1.8) | 62.3 (5.6) | 5388 | 23773 | 9911 | 82.45 |
| - low reasoning | 47.4 (1.3) | 65.9 (4.3) | 60.4 (1.8) | 77.7 (2.7) | 62.9 (1.9) | 60.9 (2.0) | 63.0 (2.3) | 59.4 (6.0) | 5328 | 22994 | 4860 | 49.50 |
| - minimal reasoning | 51.6 (2.4) | 56.8 (2.3) | 61.1 (3.5) | 75.0 (1.9) | 61.1 (1.3) | 59.8 (1.9) | 61.3 (1.8) | 55.2 (7.1) | 7294 | 23596 | 2282 | 33.48 |
| - high verbosity | 49.8 (3.0) | 71.1 (3.1) | 65.6 (2.3) | 78.7 (2.3) | 66.3 (1.4) | 66.0 (2.4) | 66.7 (1.7) | 62.1 (4.3) | 7133 | 23652 | 10784 | 87.94 |
| - medium verbosity (default) | 49.9 (2.0) | 69.0 (3.4) | 63.8 (2.7) | 78.0 (2.7) | 65.2 (1.8) | 63.7 (2.8) | 65.6 (1.8) | 62.3 (5.6) | 5388 | 23773 | 9911 | 82.45 |
| - low verbosity | 42.9 (2.4) | 66.4 (2.9) | 60.7 (2.9) | 78.7 (2.4) | 62.2 (1.1) | 60.9 (2.4) | 62.4 (0.9) | 60.4 (5.5) | 3732 | 23613 | 8899 | 75.84 |

**Overall Top-performing Model**   Overall, GPT-5 achieves the best performance in Tab. 3 at 65.9%, confirming the benchmark's challenging nature compared to popular benchmarks like AIME 25 (GPT-5 reaches 94.6%), GPQA-Diamond (GPT-5 reaches 87.0%) and SWEBench Verified (GPT-5 reaches 72.4%) (OpenAI, 2025). The result shows that ProfBench is approximately as challenging as HealthBench, where GPT-5 reaches 67.2%, despite covering many diverse domains and cheaper evaluation cost (*e.g.,* one round of HealthBench evaluation takes up to $300 with o3 while the same model on ProfBench only costs $48). Among the domains, Physics is most challenging (49.3%), followed by Finance (63.8%), Chemistry (70.6%) and Consulting (80.0%). We further analyze performance across domains in §K and the best performing model at each price-point in §F.

**Are Closed-source Models Better than Open-weight Models?** In general, the top-performing models tend to be proprietary models such as GPT-5 (65.9%), o3 (61.4%) and Gemini 2.5 Pro (60.3%). The top open-weight models are GPT-OSS-120b performing at 54.9% and DeepSeek V3.1 (Thinking) at 53.8%. The performance gap between closed-source and open-weight models is small for domains like Physics (<1%), moderate for Chemistry and Consulting (9.2% and 9.6%) and particularly large for Finance (15.0%). This might be a result of open-weight models having more in-domain training data and potentially over-emphasizing benchmarks relating to Code (e.g. LiveCodeBench) and Math (e.g. AIME 25) which are similar in problem-solving approach to Physics. On the other hand, much less attention has been placed on measuring and improving model performance in Chemistry, Consulting and Finance. We believe ProfBench can facilitate measurement and thereby catalyze progress in model capabilities within these domains, especially among open-weight models.

**Does Model Size Matter?** Similar to LLM-Judge performance, larger models tend to perform better within each model family. However, size alone exhibits diminishing return across the model families—for instance, GPT-5-mini shows a 10.2% improvement over GPT-5-nano, but GPT-5 only shows a 5.6% gain over GPT-5-mini. A similar trend is also observed with the Gemini-2.5 family (Thinking), as well as open-weight model families such as Llama 4 and Qwen3-Instruct-2507. This suggests that improvement in ProfBench necessitates not only model scaling, but possibly further innovations in training techniques and data curation.

**To Think or Not to Think?** Using the same model for which thinking can be either enabled or disabled, turning on the thinking feature slightly improves overall performance (between 0.3 to 2.3%) as exemplified by Gemini-2.5-Flash (and its Lite sibling), Claude-Sonnet-4 and DeepSeek V3.1. Similarly, increasing reasoning effort of GPT-5 from minimal to high gradually increases overall performance by 4.8%. However, when inferring with separate models of identical size trained for instruction following and thinking respectively, thinking does not necessarily confer an advantage. For instance, Qwen3-30B-A3B-Thinking-2507 scores 44.6% while Qwen3-30B-A3B-Instruct-2507 records 49.3% in overall performance, as similar to their larger 235B cousins. This might be because the instruct version generates much longer response (11167 characters on average) compared to the thinking version (4757 characters on average), which we investigate below.

**Is there any Advantage for Longer Response Lengths?** Intuitively, longer responses tend to cover more content and hence increase the chance of satisfying more criteria. This explains the reason behind the poor performance of Claude-3.5-Haiku (27.6 %), which generates only 1784 characters on average, less than half of the next most concise model. However, we also find that beyond a certain threshold, longer responses do not warrant better performance. For instance, o3 scores 61.4% with only an average of 4158 characters while GPT-5-nano scores 50.1% despite having more than twice the average response length (9796 characters). To better understand the effects that response verbosity plays in influencing ProfBench performance, we experiment with changing the verbosity flag on GPT-5. With low verbosity setting, the overall score dropped by only 3.0% compared to medium verbosity, even though the average response length dropped by 30.7%. Similarly, with high verbosity, the score increases by 1.1% compared to medium verbosity even though the average response length increased by 32.4%. This suggests that while response length does influence performance, its effect is minimal - typically within the standard deviation of the two verbosity settings.

**Can Inference Cost be Further Reduced without Affecting Robustness?** With o3, running ProfBench with 16 responses per task costs $48, which is substantially cheaper than HealthBench or PaperBench at $300 and $8000 respectively. Here, we further explore if ProfBench can be even more accessible by cutting down the cost, without compromising the robustness of the evaluation. To address this, we first observe that the performance variance differs significantly across the 40 task evaluated (Fig. 3)—for example, the score of Gemini-2.5-Flash on Task Chem-9 ranges from 11.2 to 63.8, whereas its score on Task Chem-4 remains stable, ranging from 82.0 to 96.1. This suggests that allocating a fixed number of samples uniformly across all tasks may be suboptimal; instead, the overall variance can be reduced without additional cost by generating more samples for high variance tasks and fewer for low variance tasks. We formulate this as an optimal allocation problem in integer programming, where a fixed budget (*i.e.,* number of generations) is distributed across tasks to minimize the variance of the overall performance. Given the small scale of the problem, we solve it efficiently using dynamic programming. For formal description of the problem and comparison

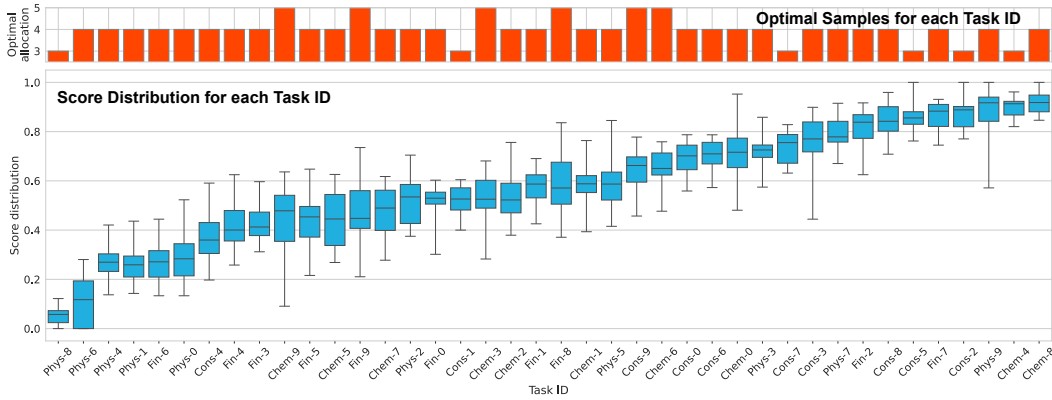

Figure 3: Score Distribution and Optimal Samples for various tasks. Each blue box represents 25th, 50th and 75th percentile and whiskers represent the worst and best score out of 16 samples for a task by Gemini-2.5-Flash (Thinking). Tasks with lower variance can be estimated with fewer samples (shown as height of orange bar), reducing inference cost w/o sacrificing estimation robustness.

with baseline methods, see §G. This allows us to allocate an average of 4 responses per tasks (and cut the inference cost to 25% of the original) without compromising robustness of our estimations.

Table 4: Ablation of LLM as report-generators. Higher is better for Score and Accuracy.

| | Score | | | | | Accuracy by Criterion Type | | | Length | Tokens | | |
|---|---|---|---|---|---|---|---|---|---|---|---|---|
| Model | Phys | Chem | Finance | Consulting | Overall | Extract/Recall | Reasoning | Style | Chars. | In | Out | $ |
| OpenAI/o3 | | | | | | | | | | | | |
| + LLM only | 39.9 (1.9) | 55.4 (3.5) | 43.4 (2.3) | 69.2 (2.8) | 52.0 (1.5) | 43.3 (1.8) | 54.8 (1.8) | 63.1 (6.6) | 4084 | 467 | 5383 | 28.16 |
| + search capability | 40.9 (2.6) | 55.8 (3.6) | 55.8 (3.9) | 71.9 (1.7) | 56.1 (1.2) | 51.1 (1.9) | 57.9 (1.6) | 60.8 (5.4) | 4438 | 242370 | 5946 | 340.68 |
| + grounding documents | 46.1 (2.8) | 61.8 (2.2) | 60.9 (2.8) | 76.8 (1.9) | 61.4 (1.3) | 60.4 (1.9) | 61.8 (1.3) | 63.0 (5.0) | 4158 | 18445 | 4709 | 47.72 |
| OpenAI/o4-mini | | | | | | | | | | | | |
| + LLM only | 36.8 (1.7) | 49.4 (2.9) | 33.7 (3.1) | 65.3 (2.6) | 46.3 (1.2) | 37.1 (2.3) | 49.4 (1.4) | 63.4 (6.3) | 3232 | 467 | 4293 | 12.42 |
| + search capability | 39.1 (2.6) | 54.7 (3.5) | 50.0 (2.7) | 69.2 (2.8) | 53.3 (1.7) | 48.4 (2.4) | 54.0 (1.7) | 60.4 (5.6) | 4774 | 127403 | 6409 | 107.74 |
| + grounding documents | 45.5 (1.9) | 58.5 (2.2) | 54.7 (2.5) | 74.4 (2.7) | 58.2 (1.0) | 55.8 (1.7) | 58.3 (1.8) | 61.0 (5.6) | 3886 | 31679 | 4763 | 35.71 |

# 6 ABLATION: HOW IMPORTANT ARE GROUNDING DOCUMENTS?

**Task Formulation** Recent benchmarks such as Humanity's Last Exam report that model performance can be significantly improved via web search (xAI, 2025; OpenAI, 2025). To understand the effect of web search and the grounding documents in each task, we conduct two ablations—(1) prompting without the documents, and (2) prompting without the documents but allowing the models to retrieve relevant documents through web search (as all documents have publicly accessible urls).

**Results** With both o3 and o4-mini, removing grounding documents greatly reduces the performance (9.4 to 11.9 %), suggesting that explicit reference to the documents is crucial for the generation quality; the effect is particularly pronounced in information extraction / recall (17.1 to 18.7%). We notice that without grounding prompts in such documents, models commonly respond with clarifying requests/questions (*e.g., Please supply the REIT's Q1'25 NOI, total assets, total liabilities, shares outstanding and 3-month ADV ...*), especially for finance and consulting. Adding search capabilities recovers some performance (4.1 to 7.0%), indicating that web search can identify some relevant documents. However, the large amount of input tokens (0.12 to 0.24 million per task, or 4 to 12x as much as the original grounding documents) suggests that a large quantity of document tokens might be included in prompt context, highlighting retrieval precision as a potential area for improvement.

# 7 CONCLUSION

We present ProfBench, the first benchmark with expert-curated rubrics across diverse professional domains including Physics PhD, Chemistry PhD, Finance MBA and Consulting MBA. ProfBench addresses a core limitation of existing benchmarks by moving beyond exam-style tasks with short answers, enabling systematic evaluation on open-ended problems with real-world value. ProfBench is also fair and accessible with reduced LLM-Judge bias and substantially lowered evaluation costs.

ETHICS STATEMENT

All data collection carried out on this project was performed by our vendor, following internal reviews on ethical and legal standards prior to the start of the project. All individuals engaged in the project were qualified for their respective roles and provided services under fair and appropriate working conditions. Compensation for all personnel was set at or above locally applicable standards, and no practices involving coercion, overwork, or unfair treatment were employed. Prior to the inclusion into the project, annotators sign an agreement not to infringe on any third party's intellectual property or other proprietary rights, indemnifying our vendor if any such claims were brought up by third parties.

REPRODUCIBILITY STATEMENT

Procedures for data collection has been extensively documented in §3, C and D. Evaluation details are in §4.1, 5 and B.

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

## A  EXAMPLES

---

**ProfBench Chemistry PhD Example**

Acid-base reactions in commercial applications such as electroplating often involve multiple coupled equilibria that must be regulated in real time to maintain desired ratios of concentrations in a mixture. Help me with the following calculations for the titration of a 100 mL mixture of acetic acid (0.5 M) and formic acid (0.1 M) with 0.5 M NaOH:
1) Calculate the volume of NaOH titrant required to reach the point where the two conjugate bases have equal concentrations.
2) Calculate the concentrations of the acids and their conjugate bases at the point referenced in part 1.
3) Calculate the concentration of hydronium ions and the pH of the analyte at the point referenced in part 1.
4) Calculate the volume of NaOH titrant required to reach the point where the pH of the analyte is 7.0.
5) Calculate the concentrations of the acids and their conjugate bases at the point referenced in part 4.
6) Calculate the volume of NaOH titrant required to neutralize both acids.
7) Calculate the concentrations of the acids and their conjugate bases at the point referenced in part 6.
8) Calculate the concentration of hydronium ion and the pH of the analyte at the point referenced in part 6.
*Example Extraction Rubric*: Determines the volume of NaOH titrant required to reach the point where the pH of the analyte is 7.0 as 0.11938 ± 0.001 L
*Example Reasoning Rubric*: Determines the pH of the analyte at the point at which both acids are neutralized as 9.05±0.05.
*Example Style Rubric (another task)*: The molecular weight is rounded to 1 decimal place.

---

**ProfBench Physics PhD Example**

In a variant of a KSVZ-style axion model, five heavy vector-like fermions are introduced, each with different Peccei-Quinn charges for left- and right-handed components $(X_L, X_R)$. Their Standard Model gauge quantum numbers and PQ charges are:
1. $Q_1 : (\mathbf{3}, \mathbf{2}, +\frac{1}{6})$, $X_L = +1$, $X_R = 0$
2. $Q_2 : (\mathbf{6}, \mathbf{1}, -\frac{1}{3})$, $X_L = +\frac{1}{2}$, $X_R = -\frac{1}{2}$
3. $Q_3 : (\overline{\mathbf{3}}, \mathbf{3}, +\frac{2}{3})$, $X_L = +1$, $X_R = +1$
4. $Q_4 : (\mathbf{8}, \mathbf{2}, +\frac{1}{2})$, $X_L = +\frac{3}{2}$, $X_R = 0$
5. $Q_5 : (\mathbf{10}, \mathbf{1}, -1)$, $X_L = +1$, $X_R = +\frac{1}{2}$
Here $Q$ is the electric charge $Q = T_3 + Y$, where $T_3$ is the third component of weak isospin and $Y$ is the hypercharge, and we work entirely in Standard Model hypercharge normalization. $T(r)$ denotes the SU(3) Dynkin index and $d(r)$ the representation dimension. For the SU(3) representation $\mathbf{10}$, take $T(\mathbf{10}) = \frac{15}{2}$.
Using the chiral PQ charge difference $(X_L - X_R)$ as the weight for each Weyl fermion, compute the ratio $E/N$ (electromagnetic to color anomaly coefficients) and give your answer in lowest-term fractional form.
*Example Reasoning Rubric*: Sums the individual color anomaly contributions to obtain the total color anomaly $N = N_1 + N_2 + N_3 + N_4 + N_5 = \frac{65}{4}$.
*Example Extraction Rubric*: Calculates $w_5 = 1 - \frac{1}{2} = \frac{1}{2}$ using $X_{L,5} = 1$ and $X_{R,5} = \frac{1}{2}$.
*Example Style Rubric*: Calculates the ratio $E/N = (\frac{58}{3})/(\frac{65}{4}) = \frac{232}{195}$.

---

**ProfBench Consulting MBA Example**

ABC Education (the "Client") delivers premium after-school STEM in English for ages 6-11 and is evaluating a partner-led Hong Kong entry. The entry would involve its renting rooms from international schools and using part-time Native English Teachers (NETs).

Formatting: (i) British English, (ii) No tables, (iii) Every HKD figure must be prefixed HK$, (iv) Show intermediate steps when calculating quantitative answers rather than outputting formulas, and (v) Round HKD only at the end of each sub-task (i.e., do not round figures for intermediate calculations, unless explicitly stated).

Task 1: Competitor dynamics ($<=100$ words) ABC Education's main competitors are (i) Big Bang Academy, (ii) Blueinno, and (iii) ESF Glenealy School. For each competitor, state the following: 1. Pricing: Output each competitor's class length + class count + package price in the following format: (e.g., "4 classes - 60 min per class - HK$4,360"). Compute HK$/h (round at the end). If length is missing, write "Cannot compute (no per-hour figure)". 2. Delivery: Pick one (i) On-Campus Partner (classes hosted at partner school premises), (ii) Learning Center (dedicated provider-run teaching location), or (iii) Hybrid: Kit + Video (take-home kit plus guided videos). 3. Pedagogy: Pick one (i) Teacher-Centered (teacher leads instruction; students follow), (ii) Project-Based (students build projects to learn concepts), or (iii) Inquiry-Based (students investigate questions; teacher facilitates).

Task 2: Hourly economics ($<=160$ words) The Client aims to run a series of courses. They wish to understand the hourly economics of their prospective endeavor in Hong Kong. For the next questions, assume the following about the Client's program: - Tuition price per class hour is HK$390; capacity 8 students per class. - As a first step, calculate paid seats assuming they are equal to 85% of capacity; the remaining 15% will be free to offer scholarships and incentives to drive demand. - Apply a 15% sibling discount to 25% of the paid seats. - Apply a 2% leakage margin of safety to the post-discount tuition. This will result in Net Tuition. - Venue rental fee: The Client has a global venue rental partner who is offering HK$500 per class hour for venue rental, but the Client does not wish for the venue rental cost to be higher than 22% of Net Tuition (which is standard for the countries in which the Client operates). As such, choose the higher of HK$500 or 22% of Net Tuition for the venue rental cost. - NET labor: Assume HK$480 per class hour teaching plus half an hour of prep at HK$240 per class hour. - Processing Fee: 5% of Net Tuition plus HK$5 multiplied by the paid seat count. Now solve the following: 4. Compute Net Tuition, Direct costs (venue + NET labor + processing), and contribution margin per class-hour. 5. State whether 22% of Net Tuition exceeds HK$500. 6. Monthly profit: Determine the monthly profit if the Client ran two classes per week across four weeks. 7. Identify whether ABC Education's tuition fee is either (i) higher than all competitors, (ii) in between competitors, or (iii) lower than all competitors.

Task 3: Venue rental fit ($<=100$ words) Harrow and DSC have approached the Client offering venue rental services. The Client wishes to understand what they are offering and whether they should partner with either school as opposed to their existing global venue rental partner. 8. Venue rental offers: Harrow is offering weekend windows at their standard base rental rate for the first hour, then HK$150/h thereafter, +10% weekend surcharge. DSC is offering weekday windows at their standard base rental rate for two hours, + HK$500 tech, + HK$200 cleaning per booking. Calculate the total venue rental cost per school, assuming the Client is looking for two-hour windows. 9. Comparison vs. global venue rental partner: Compare each school's offered rate vs. the global venue rental partner's rate. State which provider the Client should partner with (i.e., Harrow, DSC, or the existing global venue rental partner).

***Example Reasoning Rubric:*** Calculates the processing fee by multiplying net tuition by 5% plus paid seats multiplied by HK$5.

***Example Extraction Rubric:*** Classify ESF Glenealy pedagogy as Inquiry-Based

***Example Style Rubric:*** Keep Task 3 within 100 words total.

**ProfBench Finance MBA Example**

You are helping me assess the potential for a new business unit within a major investment bank. This unit is entirely dedicated to innovative finance for healthcare, as well as social impact and environmental challenges at the global level. One example we are studying to assess this opportunity is how the Global Alliance for Vaccination and Immunization (GAVI) has been able to raise money on capital markets through the International Finance Facility for Immunization (IFFIm).
I would like you to help answer the following questions:
- How was IFFIm able to raise money on capital markets for vaccination and immunization campaigns?
- How did IFFIm apply securitization?
- What were the factors that made it possible for IFFIm to raise money on capital markets?
- What risks and challenges does IFFIm have to address and overcome?
- How would you assess whether IFFIm has been effective and successful at raising funding for GAVI, and overall, has it been a success for their operations and health goals?
- Can IFFIm be viewed as a blueprint for investing in and funding other areas of global health, or other major social or environmental challenges? If yes, identify a short pipeline of 3 to 5 organizations/topics that could take a similar approach and use innovative finance to raise funding to advance their goals.
Provide a robust context, starting with defining what Gavi and IFFIm are, how they work, and how they are both related. Discuss, in detail, the technical aspects of how IFFIm works, and describe what makes it possible and effective. The technical discussion should describe how it works, and provide a detailed overview of how it raised money until now (this should include a summary of past issuances, showing on what markets it raised and what investors subscribed). Also, discuss what elements made IFFIm possible, as well as what risks and challenges IFFIm faces. This should help you form a robust and documented view on what makes IFFIm a success (or not), and how it could act as a blueprint (or not) for other initiatives.
Regarding format, I am looking for the style of a detailed investment memo discussion. You can use tables and bullets, but the memo should be mostly text, and you should walk me through your reasoning. I don't want the output to be mostly tables and bullets.
*Extraction Rubric Sample*: States that a breach of IFFIm's liquidity policy could negatively impact IFFIm's rating profile.
*Reasoning Rubric Sample*: States that vaccines are one of the most successful and cost-effective health investments in the world.
*Style Rubric Sample (another task)*: Present findings clearly to allow for effective use.

## B  PROMPT TEMPLATES

**Main prompt template**    `Response: <response>`
`Evaluate whether the response above satisfies this criterion:`
`<criterion_description>. Only answer Yes or No.`

**Alternative prompt template 1**    `Response: <response>`
`Evaluate whether the response above satisfies this criterion:`
`<criterion_description>. Only answer True or False.`

**Alternative prompt template 2**    `Response: <response>`
`Evaluate whether the response above satisfies this criterion:`
`<criterion_description>. Only answer 1 or 0.`

**Further Discussion on Prompt Templates**    We found the main prompt work substantially better than the alternative 1 (by 2.2%) or alternative 2 (by 4.5%). We believe this is because Yes/No are more commonly used in natural language compared to others. We also attempted more elaborate prompt templates (e.g. informing the LLM that they are an expert in grading responses, giving specific instructions to analyze the entire solution from every angle, providing reasoning process).

However, we did not observe any gains and had substantially increased token expenses. Therefore, we used our main prompt template, while noting that reasoning LLMs can still do reasoning within their thinking traces without explicit prompting.

## C  ANNOTATOR RECRUITMENT

**Annotator Countries**

1. United States: 20
2. United Kingdom: 5
3. Canada: 4
4. India: 4
5. Australia: 2
6. Spain: 1
7. Greece: 1
8. France: 1

**Consulting MBA:** We generally require annotators to have had 2 years of work experience at McKinsey, Boston Consulting Group, Bain & Company, Deloitte, PricewaterhouseCoopers, Ernst and Young or KPMG. Alternatively, they could have 4 years of experience at another consulting firm. These work experience includes those prior to the completion of their highest degree (i.e. MBA). Candidates are interviewed based on their past experience, their professional ability to reason from first principles, communicate in a professional manner, break down projects into logical components, and create well-supported recommendations among others.

**Finance MBA:** We generally require annotators to have 2 years of experience at a select bank including JPMorgan, Bank of America, CitiGroup, Wells Fargo, Goldman Sachs, Morgan Stanley, PNC, Truist, TD Bank and other banks with similar selectivity in regions outside of North America. These work experience includes those prior to the completion of their highest degree (i.e. MBA). Candidates are then interviewed on their previous experiences - projects, deals, investments, credits, or portfolio decisions including their individual contributions, decisions made, and alternatives considered while ensuring applicants discuss key risks, explain context, and suggest mitigations.

**Chemistry and Physics PhD:** We generally require annotators to have completed a PhD relating to Physics or Chemistry at a Global Top 100 university while taking into account other factors such as the program selectivity. Candidates are then evaluated based on interviews that test how deeply they understand their field, their ability to synthesize new information related to their niche, and clear communication of findings.

## D  FURTHER DESCRIPTIVE STATISTICS

**Prompts**  ProfBench contains 80 unique prompts, with 20 from each domain. Prompts are generally around 2 to 3 paragraphs long with an average of 2052.4 characters (std of 997.4, min of 696, max of 5339). As shown in the examples (see §A), prompts typically contain multiple related sub-questions, making them substantially more specific to construct evaluation criteria for and therefore minimizing the accidental penalization of responses that approach the task in alternative ways. Chemistry and Physics PhD tasks tend to be shorter averaging 1617.4 (std of 925.6) and 1799.2 (std of 622.9) characters respectively while Consulting and Finance MBA tasks tend to be longer at 2360.8 (std of 717.2) and 2432.2 (std of 1314.3) characters respectively.

**Responses**  are generated with OpenAI o3, DeepSeek R1-0528 and Grok4 models. They are similar in length, with o3 being the most terse at 4816.7 characters (std of 2590.5), followed by R1-0528 at 5187.7 (std of 2504.2) and Grok4 is most verbose at 5997.2 (std of 3789.3) characters. Across the different models, responses to Physics PhD are generally most succinct at 3839.5 characters (std of 2319.0), followed by Finance MBA at 4924.8 characters (std of 3383.0), Chemistry PhD at 5609.3 characters (std of 2934.7) and finally Consulting MBA at 6961.8 characters (std of 2620.0).

Table 5: Rubric taxonomy with sub-categories, and representative examples. Reasoning dominates with the majority of checks on logical validity and causal correctness, while Extraction emphasizes faithful and detailed retrieval, and Style focuses primarily on formatting and clarity.

| Category | Concise Description | Representative Example |
|---|---|---|
| **Extraction (Recall)** | Evaluates whether the model retrieves the right information with correct coverage, granularity, and faithfulness. | "Identifies RTX as one of the four companies with the largest DoD obligations to the US government in 2022." |
| **Reasoning** | | |
| Logical Validity | Assesses whether reasoning steps are logically sound, consistent, and free of contradictions. | "Reasons that Anduril could have higher gross margins due to its business model because software has higher margins than hardware at scale." |
| Causal/Mathematical Correctness | Applies correct formulas, computations, and causal inferences. | "Calculates 2020–2024 defense spend CAGR by dividing 2024 spending by 2020 spending, raising to the power of 1/4, then subtracting 1." |
| Completeness of Reasoning | Shows intermediate steps or justifications rather than jumping to a conclusion. | "Shows steps for intermediate calculations when arriving at quantitative answers rather than quoting formulas." |
| Generalization & Abstraction | Combines information or draws higher-level insights beyond literal recall. | "Notes that strong donor support, a robust risk framework, and proven impact are necessary to replicate the IFFIm model." |
| **Style** | | |
| Clarity & Readability | Organizes response clearly and makes it easy to follow. | "Displays answers and includes a concise summary tying together key findings: price change, EPS impact, valuation change." |
| Conciseness | Avoids verbosity and keeps the response tight and relevant. | "Keep Task 2 within 160 words total." |
| Formatting | Correctly follows requested structure, notation, and units, grammar and spelling. | "Quotes all percentages to one decimal place." |
| Tone & Appropriateness | Matches the expected tone (formal, neutral, or explanatory). | "Presents analysis in structured consulting framework format." |

**Rubrics**    Each task comes with 15 to 59 individually-gradable criteria, with a mean of 30.6 and std of 9.9. Chemistry has the fewest criteria at average of 26.0 followed by Consulting (30.5), Finance (32.9) and Physics (33.0). In terms of rubric weight, around half (49.8%) fall under Major, with roughly a quarter in Critical (23.4%) and Minor (23.9%) and a tiny fraction in Additional (2.9%). Our analysis shows that in terms of criterion types, Reasoning dominates, covering 62.9 % of all items. This category primarily assesses whether the model's reasoning process is sound, complete, and coherent. Extraction (Recall) accounts for 34.1% of items, focusing on whether the model retrieves the correct information with sufficient coverage and granularity. Style makes up the remaining 3.0%, evaluating how well responses are communicated in terms of clarity, structure, formatting, and tone. This suggests that ProfBench emphasizes on extracting and reasoning with professional knowledge, and less so on the stylistic presentation of the response. To provide further insight, we divide Reasoning and Style into sub-categories by collaborating interactively with Qwen3-235B-A22B-Instruct-2507. Specifically, we prompted the model to come up with initial sub-category candidates based on a set of randomly selected criteria within the Reasoning and Style categories. Then, the research team vetted the proposed candidates to make sure that the sub-categories are non-overlapping and at a suitable granularity. Once the research team is confident about the subcategories, we prompt the model to individually classify each criterion based on the concise definitions in Tab. 5. Our results are in Fig. 2, with a representative sample for each sub-category in Tab. 5 . Extraction (Recall) is kept as a single category since its core criterion is straightforward. Within Reasoning, most checks target logical validity (over 70%), with a smaller but notable portion addressing mathematical or causal correctness (about 24%). Within Style, the majority of items focus on formatting (over 70%), followed by clarity and readability (about 19%).

**Response Annotation**    Overall, o3 fulfills 51.6% of criteria as annotated by humans, while Grok4 fulfills 47.4% and R1-0528 fulfills 45.2%, indicating a relatively small gap between SOTA closed-source and open-source LLMs. Analyzing across domains, the three model responses on average fulfills only 39.1% of criteria in Finance and 40.3% in Physics, suggesting that they are harder domains for SOTA LLMs. Conversely, Consulting criteria are fulfilled 56.5% and Chemistry criteria are fulfilled 59.4% of times, suggesting that they contain relatively easier tasks for SOTA LLMs.

**Grounding Documents**    During our initial collection, annotators are requested to identify CSV or PDF documents from the public internet, which can support answering the task. However, this resulted in some documents being as long as 838 pages, with 200-400 page documents being relatively common. Each file has an average of 42.3 pages (std of 86.5), with the sum of files across each task

reaching 141.7 pages (std of 193.5). After running some initial experiments, we found that such documents often cannot fit into the limited context windows of popular LLMs. For instance, OpenAI o3/o4-mini and Anthropic Claude-Sonnet 4 only supported up to 200k context tokens while many open-weights models (e.g. GPT-oss-120B or DeepSeek V3.1) only support 128K context tokens. To ensure that ProfBench is compatible with these LLMs, we ask annotators to perform an additional step to truncate these documents by identify the most relevant information: Each pdf file is no longer than 20 pages and each csv file has less than 100 rows and 10 columns. Post-truncation, this was reduced to 7.37 pages per file (std of 3.10, max of 15) and 24.69 pages per task (std of 19.28, max of 85, min of 3). This truncation step will make tasks substantially *easier* but we believe this is a necessary step in order to make benchmarking many current models *feasible*. As the average context length increase in the future, we can use the original documents directly, which will serve as a better, realistic use of long context evaluation in the future.

In addition, some LLM providers such as OpenAI restrict the max size of each file to 10 MB, with a total of no more than 32MB across all files for a request. Therefore, we implement additional requirements for annotators to restrict each file to 10 MB, up to 10 files per task and 30MB across all files for a task. We also noticed that many popular providers' API (e.g. OpenAI and Google) can natively process PDF documents but not CSV documents, therefore requiring them to be passed as either plain text or processed through a code interpreter. Given that there were only 4 CSV files in total across ProfBench, we decided that the best workaround was to convert CSV documents into PDFs showing the same data, using pandas and pdfkit (JazzCore, 2025). These resulting PDFs contain table headers on every page, allowing rapid identification of relevant information. Each task contains an average of 3.35 files (std of 2.71, max of 10, min of 1). The size of each file is 0.813 MB (std of 1.196, max of 9.165, min of 0.017). The average size of all files in each task is 2.723 MB (std of 2.92, max of 12.555, min of 0.085).

## E    INFERENCE SETUP

**Response Annotation during Data Collection**    Temperature were set at 0.2 (except o3, which doesn't allow setting temperature and uses medium effort), with access to a web search tool. o3 and Grok use a native search tool while DeepSeek model uses SearXNG (2025) and document upload support (pdf documents were uploaded natively while csv files were uploaded as plain text). Note that responses were generated with the un-truncated documents at this stage.

**Report Generation**    Google models use default temperature of 1 as we observe that temperature 0 induces highly repetitive generations and Kimi-K2-0911 uses the recommended inference temperature of 0.6, similar to reasoning models. Note that these are slightly different from Response Annotation during Data Collection inference setup, since those were run by our vendor. Despite the relative affordability of the judge, we incur approximately \$3.50 for judging criterion-fulfillment alone (outside of response generation). We use the native PDF documents processing capabilities for OpenAI and Google models, and use OpenRouter's default document processing for other models. For models using OpenRouter document processing, cost does not include approximately \$1 charged by OpenRouter for PDF processing (when using file annotations to 'cache' file processing).

**Difference between LLM-Judge performance and Human Grading Performing of three models** R1-0528 - pred 46.8 vs human 46.1; Grok4 - pred 50.5 vs human 51.8; o3 - pred 53.5 vs human 52.7. Note these score are not comparable to those in §5 as they were generated by our vendor as discussed above.

**Judge Sensitivity to Response Generation Model**    We selected the three representative models (o3, Grok4 and R1-0528) intentionally as they represent the top-performing systems at the time of data collection (July 2025), spanning diverse providers and paradigms: two proprietary (o3 from OpenAI and Grok4 from xAI) and one open-weight (R1-0528 from DeepSeek). Ideally, sensitivity analysis would involve re-annotating responses from alternative reference sets with human experts, but this is resource-intensive due to the need for PhD/MBA-level judgments on thousands of new response-criterion pairs. Nevertheless, our existing analyses (including post-submission checks) demonstrate that the evaluations are robust and unlikely to change materially with different references:

As detailed in Section 4.2 and Table 2, top LLM-Judges (e.g., Gemini-2.5-Pro at 78.2%, gpt-oss-120b at 78.2%) exhibit extremely low bias-index values (less than 0.5% across these three reference models), meaning they align closely with human annotations regardless of the response's origin. This low spread suggests that good judges are impartial to the specific mix of proprietary/open-weight styles in our set. The impartiality of the LLM judges in ProfBench is in a large part contributed by the nature of the evaluation to decompose the LLM judgement into 15-60 rubric criteria per prompt that we independently score, then weighted together. Our approach results in substantially lower bias (in both relative terms $< 0.5\%$ and absolute terms across 3 models $< 1.0\%$ for our official gpt-oss-120B judge) compared to alternative approaches of applying LLM judges (e.g. picking preference among two responses (Dubois et al., 2023; Li et al., 2024; Zheng et al., 2023) or giving a single scalar score for each response (Zheng et al., 2023)) where biases can be as high as 30 to 75%.

To directly test the null hypothesis that judge evaluations do not depend on the specific reference model identities, we conduct a permutation test (10,000 iterations) on judge evaluation. Specifically, we fix the judge predictions and human labels, randomly shuffle the model labels assigned to responses, and recompute the Bias Index under each shuffled labeling. For our final judge model, gpt-oss-120b, the observed Bias Index falls well within the null distribution, with two-sided p-value ¿ 0.56. This confirms there's no evidence that the evaluation depends on the specific reference model; the low bias and high agreement we report are genuine properties of the judges, rather than the artifacts of our particular reference set.

**Ablation**   Cost excludes search query costs, which is is estimated to be around $6.40.

## F  OPTIMAL PERFORMANCE ON PROFBENCH AT EACH PRICE-POINT

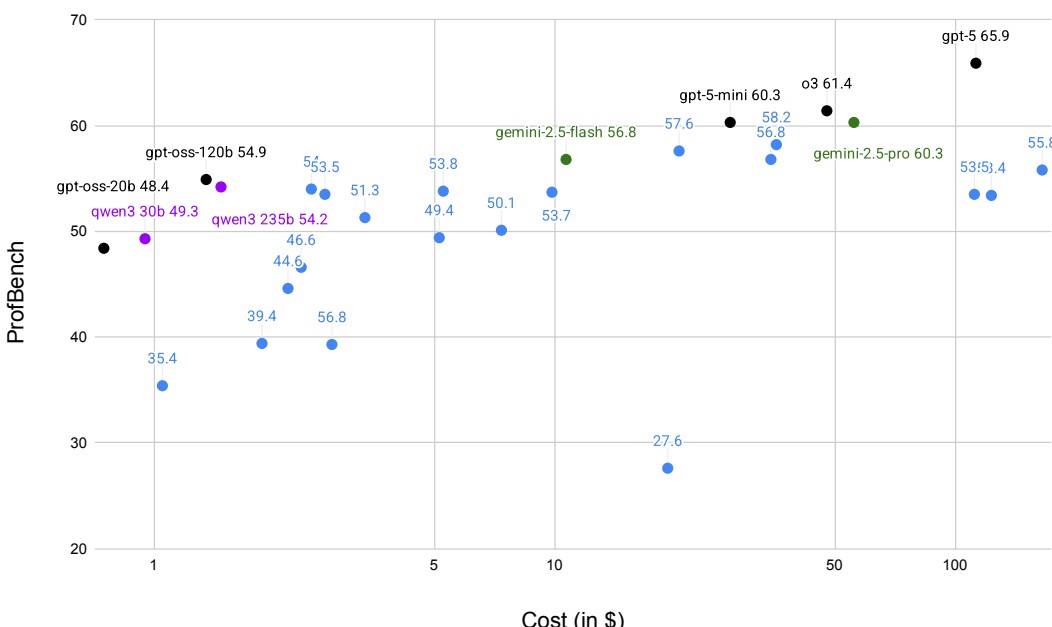

Figure 4: Optimal ProfBench performance at each price-point. OpenAI models are on the Pareto Frontier at each price-point, likely because of first-mover advantages in rubric-style data such as HealthBench (Arora et al., 2025) and PaperBench (Starace et al., 2025). Gemini-2.5 and Qwen3-Instruct-2507 models are close to the Pareto Frontier.

## G  FORMULATING OPTIMAL ALLOCATION PROBLEM

As exemplified in Fig. 3, we observe that the variance of model performance differs significantly by task. Motivated by this finding, we propose to reduce the variance of the estimated model performance by dynamically allocating different number of responses to each task —- ideally allocating more for tasks with high variance, and less for tasks with low variance.

More formally, let $n_i$ denote the number of generations allocated for task $i$. Our goal is to minimize the variance of the overall performance $S$ across $N$ tasks, while satisfying the total number of generations $\sum_i n_i$ to match a fixed budget $B$. First, the overall performance $S$ is defined as:

$$S = \frac{1}{N} \sum_{i=1}^{N} \overline{s}_i$$

where $\overline{s}_i$ is the mean score over the $n_i$ responses we generate for task $i$. Assuming i.i.d. sampling of the responses and the independence between $N$ tasks, we compute the variance of $S$ as:

$$\sigma^2(S) = \frac{1}{N^2} \sum_{i=1}^{N} \frac{v_i}{n_i}$$

where $v_i$ is the variance associated with task $i$, which we estimate by generating 16 rollouts per prompt across 4 representative models and aggregating their variance. Now, minimizing the variance $\sigma^2(S)$ is equivalent to minimizing $\sum_i \frac{v_i}{n_i}$, subject to:

$$\sum_{i}^{N} n_i = B, \, n_i \in \mathbb{Z}^+, \, n_i \geq 1 \text{ for } \forall i$$

Given the small size of the budget and number of tasks, we directly solve the objective via dynamic programming. The results of this allocation are shown in Fig. 5. We follow the allocation scheme to

sample responses from 4 representative models—Gemini-2.5-Pro , Gemini-2.5-Flash (Thinking), o3, and o4-mini—and report their average standard deviation of overall performance.

As expected, the optimal allocation consistently achieves smaller variance compared to heuristic baselines, and we set $B$ to be 160 (i.e. average $n_i = 4$) with $N = 40$, reducing down the standard deviation to only 50% of the naive point estimate.

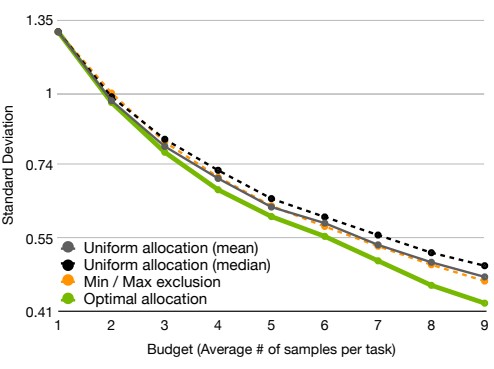

Our results, illustrated in Fig. 5, compare 4 approaches: uniform allocation (generating a fixed number of responses per task) with mean and median-based aggregation of the scores, and optimal allocation, as well as a heuristic baseline: min / max exclusion (uniform allocation followed by exclusion of min / max score responses from each task during aggregation). Overall, we find that generating multiple response per task can effectively reduce standard deviation, substantially below the level of naive point estimate (budget = 1). Furthermore, optimal allocation consistently

Figure 5: Standard deviation of overall performance using multiple samples per task. Optimal allocation of samples consistently reduce the variance across all budget levels.

reduce the performance variance compared to the alternative methods, consistently across all budget scales.

## H    FUTURE DIRECTIONS

**Tool Use**    The main tool we focus on in Section 5 is the document-ingestion tool, which allows LLMs to utilize information from PDF grounding documents that human expert annotations identify. We also conducted smaller-scale experiments with other tools such as the search tool in Section 6. We decided to de-prioritize further explorations on the search tool because the implementations of LLM search tool use are currently non-standardized. For instance, proprietary providers (e.g. OpenAI, Google and Anthropic) typically have their first-party search tool while open-weight models have to depend on third-party search tools (e.g. Exa AI). Such differences mean that when a model using search-tool is measured, it becomes challenging to understand whether model performance can be accounted for by search-tool capabilities, model capabilities or the integration of search-tools with models. In addition, we also considered and made initial attempts with other tools (e.g. code interpreter environment - which integrates spreadsheet tool and calculator tool), since we initially had a small portion of documents in CSV format. However, at the time of our experiments, only OpenAI had a code interpreter tool integrated into a publicly accessible API. Using such tools makes it challenging to fairly compare these models with a vast number of other models that do not use these tools. As the ecosystem for various tools matures, we certainly plan to incorporate further evaluations using such tools.

**Multimodality**    We purposefully designed ProfBench to be text-only (i.e. no image or audio) in order to support recent popular open-weight models since many of them are text-only models (e.g. DeepSeek V3.1 / Kimi K2 / GPT-OSS). This ensures the open-source community can readily build upon these models and utilize ProfBench to measure and improve model capabilities relevant to LLM-judge and long report generation. As the open-weight model ecosystem develops with more models supporting multi-modal capabilities, followup works can build upon the foundations laid by ProfBench.

**Other Domains**    We select the domains of Chemistry Research PhD, Physics Research PhD, Consulting MBA and Finance MBA based on three factors: a. Whether other works have covered such domains previously, b. Real-world usefulness of these domains and c. Whether we could recruit suitable annotators to robustly annotate for these fields.

For the first factor, we focused on domains that have not been previously worked on by earlier works (Arora et al., 2025; Starace et al., 2025; Guha et al., 2023; Zhou et al., 2025) in order to show fresh capabilities in new settings, which we believe to be more helpful to the community.

For the second factor, we wanted to measure LLM capabilities that real-world users care about. For instance, many existing advanced Chemistry and Physics evaluations (e.g. GPQA (Rein et al., 2023) and HLE (Phan et al., 2025)) mostly measure exam-style problems that can be answered in a Multiple-choice or a short span format. For many real-world physical scientists, answering these exam questions is less helpful in their work compared to assisting them in their research planning, as covered in ProfBench. In addition, we want to focus on some enterprise use-cases and we found that capabilities relevant to Consulting MBA and Finance MBA are relevant to many white-collar professions - meaning that measuring LLM capabilities in these domains can proxy their usefulness for many real-world professional users of AI.

For the third factor, we were constrained to annotators that we could feasibly recruit for to do these tasks within our research timeframe and budget. Given these constraints, we opted to focus on the set of 4 domains, each with a useful number of tasks (20 tasks / domain, each taking 10-20 hours of expert annotations) rather than covering a large set of domains but with a small number of tasks in each (e.g. DeepResearch-Bench (Du et al., 2025) covers 22 domains but only has an average of 2.3 tasks in each domain in English). We do not have near-term plans to expand the number of domains for newly-annotated tasks although it is possible in the medium or long term. In the short term, we recommend researchers to utilize ProfBench along other benchmarks (Arora et al., 2025; Starace et al., 2025; Guha et al., 2023; Zhou et al., 2025) where appropriate for their research goals.

**Measuring creativity in Professional Tasks**   We believe that aspects such as originality and creativity are hard to objectively measure (even for humans, they tend to be very subjective). We specifically ask annotators to phrase prompts in ways that do not necessarily require creative answers (e.g. making sure calculations are correct in a multi-step chemical equation). As models gradually become performant in objective tasks in ProfBench, we agree that the next step of evaluation might involve measuring creativity of such professional tasks (e.g. finding creative chemical experimental approaches or structuring novel financial deals), which can build upon ProfBench and prior works on measuring creativity (Lu et al., 2025).

**Risk of Saturation**   We believe that the current difficulty (50% +/- 15% for most models) is useful for many model trainers to get effective signals and improve. To reach this target difficulty, we focus on the easiest setup for report generation (Section 5), where models are given grounding documents that were truncated by human experts (to max. 20 pages each) to make it easy for LLMs to locate where the relevant information is. As models get better, we can adapt the benchmark to make it more challenging and avoid saturation. One approach is to use the original documents (up to 800+ pages long each) to better test their long context processing capabilities as many models get more than 128k context window size.

Furthermore, we can even remove the grounding documents altogether and force models to either search for relevant documents or answer prompts through parametric knowledge alone. An ablation study that we conducted in Section 6 shows that such settings cause model performance to drop by 4.9% to 11.9%. This means that once the current version of ProfBench becomes too easy for the strongest LLMs, we can easily flip a switch to make ProfBench substantially more challenging and future-proof. For instance, the top performing LLM only reaches 52% when forced to rely on parametric knowledge.

## I   REVIEW PROCESS

Review refers to the process in which the prompt and rubrics were iteratively improved. Below is the detailed description accompanied by a running example from Physics PhD pool::

1. **Annotator A submits an initial prompt**: Consider a five-level "quadpod" atom with four long-lived states $|1\rangle, |2\rangle, |3\rangle, |4\rangle$ and one excited state $|e\rangle$. In the rotating-wave frame ($\hbar = 1$), . . .

2. **Reviewer B recommends Major prompt edits**: The prompt is mathematically precise, defines all relevant symbols, and fixes a gauge, making the problem well-posed and non open-ended. It clearly states the task of computing the $3 \times 3$ non-Abelian Berry holonomy for a specified loop. For improved clarity and rubric alignment, consider: - Stating explicitly that the holonomy is to be reported in the ordered dark basis $(|d_1\rangle, |d_2\rangle, |d_3\rangle)$ evaluated at $(\alpha, \gamma) = (0, 0)$. - Noting that G is interpreted modulo $2\pi$ due to the periodic nature of $\gamma$. - Indicating whether the expected answer should be a fully simplified closed-form matrix in terms of $\theta, \phi, \chi, G$, or if an ordered product form is acceptable. - Optionally mentioning that the resulting holonomy lies in $SU(3)$ for this loop. . . .

3. **Annotator A makes changes based on Reviewer B's feedback** : Thank you for the detailed feedback! I have incorporated all the changes suggested by the reviewer. I have also changed scaffoldings and split multistep tasks into separate tasks. Hopefully the quality of the prompt has been improved.

4. **Reviewer B approves prompt**

5. **Annotator A submits an initial rubrics** : States the explicit orthonormal degenerate dark-state basis ...

6. **Reviewer B recommends minor rubric edits**: Overall, the rubric covers the full logical progression from defining the dark-state basis and gauge potentials to computing the holonomy. However, several criteria reference variables, definitions, or results from earlier criteria without reintroducing them. Each criterion should be self-contained, including all necessary definitions and expressions, so the model does not need to look back. I included my revisions for each section and checked them as well.

7. **Annotator A makes changes to rubrics** : States the explicit orthonormal degenerate dark-state basis ...

8. **Reviewer B approves rubrics**

9. **Annotator A annotates 3 model responses based on rubrics**

10. **Reviewer B recommends minor changes to model response annotations**

11. **Annotator A updates annotation 3 model responses based on rubrics**

12. **Reviewer B approves response annotations**

Annotator A is a single annotator who developed the prompt, rubrics and annotated model responses against the rubrics, as supported by Reviewer B. Note that Reviewer B is typically one reviewer but can at times be different reviewers at each stage if the original reviewer is unavailable since each task takes 10-20 hours across the three stages. Steps 3 to 4 / 7 to 8 / 11 to 12 can be repeated a few times until the reviewer is satisfied. During the rubric review process, both the annotator and reviewer are instructed to check that the rubrics fulfil a number of requirements (e.g. each can be graded in a binary fashion, should not overlap with other criteria and must collectively cover all important aspects of a good response). There is a longer list of requirements that we initially planned to include into the Appendix. However, some of this information constitutes confidential business-sensitive information for our data vendor and hence we were unable to include it into the submission.

## J SENSITIVITY AND QUALITATIVE ANALYSES OF LLM JUDGES

**Sensitivity Analysis** The ideal way to stress-test any bias metric is to compute it over a large and diverse set of reference models. In practice, however, computing the Bias-Index (or any similar fairness metric that relies on human expert ground-truth) is inherently constrained by the availability of model-generated responses but also the expert-annotated judgements for all model generations.

In ProfBench, expert annotators provided binary fulfillment judgments for responses generated by the three models that, at the time of data collection (July 2025), represented the strongest proprietary and open-weight systems on these professional tasks (OpenAI o3, xAI Grok4, and DeepSeek R1-0528). Obtaining comparable high-quality human expert judgments for many additional models would require re-running the full annotation pipeline on thousands of new generations, which is prohibitively expensive and time-consuming given the PhD/MBA-level expertise required.

Nevertheless, we believe the current three-model setup provides a sufficiently meaningful and interpretable signal of bias for the following reasons:

The three reference systems come from three distinct providers/organizations with different training data, objectives, and output styles (OpenAI, xAI, DeepSeek), providing reasonable coverage of the proprietary vs. open-weight divide that matters most for fairness concerns in the community. As shown in Table 2, the top-performing judges (gpt-oss-120b and gemini-2.5-pro) exhibit max–min differences of less than 1%, suggesting no significant preference for any single provider.

That said, we further propose complementing Bias index with two additional metrics:

*Standard deviation of biases*: the standard deviation of biases collected across 3 models. This is less sensitive to an extreme outlier than max - min while penalizing systematic spread.

*Provider-level fairness gap*: We compute the gap between average bias toward the open-weight model (R1) versus the average bias toward the two proprietary models (o3, Grok4).

Table 6: Comparison of Standard deviation of biases and Provider-level fairness gap

| Model | Standard Deviation of Biases | Provider-level Fairness Gap |
|---|---|---|
| gpt-4.1 | 0.37% | 0.65% |
| gemini-2.5-flash | 0.12% | 0.25% |
| gpt-5 | 0.51% | 0.30% |
| o3 | 0.65% | 0.89% |
| gpt-oss-120b | 0.22% | 0.15% |
| deepseek-v3.1 | 0.29% | 0.45% |

The summary of results with the new metrics are shown in Tab. 6. The top judges (as identified in our original analysis) achieve consistently small biases in these new setups, with gpt-oss-120b achieving 0.22% in standard deviation of biases and 0.15% in fairness gap. We believe these results strengthen the fairness analysis while remaining fully grounded in the high-quality expert annotations.

**Performance across categories** The overall F1 score only captures misalignment between the LLM and human experts. In addition to this, we also include the category-specific F1 (Extraction, Reasoning and Style) in Table 2. Across these categories, models generally perform better on Extraction, followed by Reasoning and finally Style. This suggests that many LLM-Judges are weaker in judging whether responses fulfil stylistic requirements (e.g. writing an investment memo in a specific format; returning chemistry calculations to the requested precision) since they are sometimes more nuances and could involve checking different parts of the response (e.g. all concentration values to two decimal places) rather than Extraction tasks that are generally more focused on a specific segment within the response.

**Qualitative Analysis** We also did a qualitative analysis of some reasoning traces, where the human annotators disagree with the optimal LLM judge (GPT-OSS-120B, high reasoning effort for Chemistry/Physics/Style, low otherwise).

Here are some patterns we observed that explains the shortcomings of the LLM-Judge:

1. **LLM Judge sometimes does not recognize the equivalence of certain statements.**
   - **Criterion:** Identifies the mechanism of the ylide formation as SN2 substitution.
   - **Explanation:** In this case, the Bimolecular (i.e., two reactants) Nucleophilic substitution (SN2) reaction produces both the ylide and the phosphonium salt, but the LLM judge wrongly focuses only on the phosphonium salt.
   - **Reasoning Trace:** The user asks: "Evaluate whether the response above satisfies this criterion: Identifies the mechanism of the ylide formation as SN2 substitution. Only answer Yes or No." We need to examine the provided response (the synthesis plan) and see if it identifies the mechanism of the ylide formation as SN2 substitution.
   The ylide formation step is the formation of the ylide from the phosphonium salt and n-BuLi. The ylide formation is actually deprotonation of the phosphonium salt by n-BuLi, which is *not* SN2 substitution but base deprotonation.

The response includes a step "Preparation of Isopropyltriphenylphosphonium Bromide," which is an SN2 substitution ($PPh_3$ attacks alkyl bromide). However, the question specifically asks about the ylide formation.

– Step 3: "Preparation of Isopropyltriphenylphosphonium Bromide: Mechanism: SN2 attack by P on alkyl halide." — This is formation of the phosphonium salt, not the ylide.

– Step 4: "Wittig Reaction to Form Dienophile: Mechanism: (a) Deprotonation by n-BuLi forms ylide. (b) Ylide attacks carbonyl..." — It does not state that the ylide formation is SN2 substitution; it is deprotonation.

Thus, the response does not identify the mechanism of ylide formation as SN2 substitution. It identifies the mechanism of phosphonium salt formation as SN2, not the ylide formation.

**Answer:** No.

2. **LLM judge sometimes mistakes implied statements for explicit statements.**

   • **Criterion:** Defines the theoretical moles of hydrazine as

   $$n_{N_2H_4} = \frac{n_{e^-}}{\text{Number of electrons}}$$

   • **Explanation:** This was not explicitly mentioned in the response but utilized as an assumption in subsequent calculations.

   • **Reasoning Trace:** We need to parse the question. The response includes a calculation: "The theoretical production rate assumes 100% Faradaic efficiency. Total moles of electrons:

   $$n_{e^-} = \frac{Q}{F} = \frac{0.23768}{96485} = 2.4633 \times 10^{-6} \text{ mol.}$$

   Theoretical moles of hydrazine:

   $$n_{N_2H_4} = \frac{n_{e^-}}{4} = \frac{2.4633 \times 10^{-6}}{4} = 6.1583 \times 10^{-7} \text{ mol.}$$

   "

   This effectively defines the theoretical moles of hydrazine as $n(N_2H_4) = n_{e^-}/4$, satisfying the criterion.

   **Answer:** Yes.

3. **LLM judge sometimes fails to integrate information from different parts of the response.**

   • **Criterion:** Proposes a substitution reaction with isopropanol and phosphorus tribromide in diethyl ether.

   • **Explanation:** The response contains two parts: (a) "All steps are designed to be carried out under standard laboratory conditions, with diethyl ether as the primary solvent" and (b) "Preparation of isopropyl bromide (iPrBr): Mix isopropanol (27 g, 0.45 mol) with phosphorus tribromide (33 g, 0.12 mol) in a round-bottom flask."

   The LLM judge missed the first part.

   • **Reasoning Trace:** The response proposes a substitution reaction between isopropanol and $PBr_3$, but does not specify diethyl ether as the solvent.

   The criterion requires the reaction to be specifically in diethyl ether.

   **Answer:** No.

**Overall**

While these shortcomings highlight that LLM-Judges are not perfect oracles, the errors tend to be systematic rather than random and therefore may have less effect on relative comparisons between models as opposed to the absolute value. As we noted in lines 331 to 332, we only saw a 0.7 to 1.3% gap between judge-predicted and human-annotated performance across 3 models that were annotated by our LLM judge and professional human annotators.

# K    REPORT-GENERATION PERFORMANCE ACROSS VARIOUS DOMAINS

Our design philosophy for ProfBench was to faithfully reflect the authentic difficulty and nature of real-world professional work in each field, rather than to artificially equalize difficulty through task selection, rubric pruning, or post-hoc reweighting. Forced equalization would risk several undesirable side effects: (i) discarding the hardest (and often most diagnostically valuable) tasks in a domain, (ii) inflating scores in inherently more complex scientific domains by relaxing rubrics, or (iii) introducing arbitrary weighting schemes that lack objective grounding and reduce transparency.

Henceforth we did not make any specific intervention on the collected data, except that we match the number of tasks in each domain to be equal. At the same time, we report the comprehensive domain-wise / reasoning type-wise performance of all models, given that the overall score - like any other aggregation methods - inevitably distills useful information about subcategory-level performances. Our design principle directly aligns with subcategories in prior benchmarks that focus on professional tasks, such as conversation theme-wise analysis in HealthBench (Arora et al., 2025) and machine-learning problem domains in PaperBench (Starace et al., 2025).

Nonetheless, we have additionally examined whether the observed absolute difficulty gaps actually distort the ranking or relative capability assessment of models when scores are aggregated. Two sets of additional analyses (conducted post-submission) show that our aggregation scheme does not harm the validity of cross-model comparisons: For each domain, we normalize all reasoning model scores by dividing with the best model score in that domain (i.e., setting the best model score = 1.0). We then compute the deviation of these normalized scores separately for each domain. In addition, we also analyze the rank correlation between the performance in each domain and the overall aggregated score.

Table 7: Standard deviation and Spearman's rank correlation across domains.

| Domain | Physics | Chemistry | Finance | Consulting |
|---|---|---|---|---|
| Standard Deviation | 0.11 | 0.10 | 0.11 | 0.09 |
| Spearman's Rank Correlation | 0.85 | 0.79 | 0.91 | 0.89 |

The resulting variances are consistently similar to each other across all 4 domains, which indicates that regardless of absolute difficulty, the performance hierarchy and gaps between strong and weak models remain stable across all four domains. In all 4 domains, the correlation marks close to or beyond 0.8, indicating very strong correlation. This directly evidences that the overall aggregated score faithfully reflects model capabilities in all domains, rather than systematically penalizing a specific subset. Taken together, these results show that our domain-based categorization and aggregation is diagnostically valid.

