# OpenReview forum: "ProfBench: Multi-Domain Rubrics requiring Professional Knowledge to Answer and Judge"
_ICLR.cc/2026/Conference — ICLR 2026 Poster_

### Official Review · Reviewer_pJFi · 2025-10-27

**Soundness:** 2
**Presentation:** 2
**Contribution:** 3
**Rating:** 6
**Confidence:** 3

**Summary:**

This paper introduces ProfBench, a high-quality evaluation benchmark spanning multiple professional domains. It contains 80 tasks and more than 7,000 human-constructed response–criterion pairs, developed entirely by domain experts with PhD or MBA degrees and without any LLM assistance, ensuring authenticity and professional rigor. The authors further propose a rubric-based LLM-as-a-Judge evaluation paradigm and design three metrics to comprehensively assess assessment consistency, fairness, and efficiency. A systematic evaluation of over 40 open-source and closed-source models investigates the influence of different reasoning mechanisms, model sizes, and response lengths, and presents strategies to reduce evaluation cost and bias. Experimental results show that even the strongest closed-source model (GPT-5-high) reaches only 65.9% on this benchmark, highlighting the professional difficulty and challenge posed by the tasks.

**Strengths:**

In the data annotation phase, the paper adopts a rigorous expert participation mechanism. A total of 38 professionals with PhD or MBA backgrounds were recruited to design tasks and formulate scoring criteria. Multiple rounds of review and consistency verification were conducted to ensure the reliability of annotations. This process guarantees the high quality of the dataset in terms of knowledge depth and annotation accuracy.

The paper systematically compares over 40 types of mainstream models, covering dimensions such as closed-source vs. open-source, different sizes, and "thinking" settings. It also analyzes the relationships between model performance, bias, output length, and reasoning costs. The overall experiments are comprehensive, and the conclusions are credible.

During the evaluation process, the authors systematically studied the performance and cost differences of different judge models, and proposed an optimal sample allocation method and a low-cost evaluation scheme. This strategy can significantly reduce evaluation costs while maintaining high consistency, which provides valuable insights for large-scale evaluation practices.

ProfBench covers four professional domains: Physics, Chemistry, Finance, and Consulting. With diverse task types, it can comprehensively reflect the generation and judgment capabilities of large language models in professional scenarios.

**Weaknesses:**

1. The paper describes the annotation process and consistency metrics in detail, but lacks qualitative demonstration of controversial samples and explanations of the adjudication mechanism. Given the subjectivity of rubric-based evaluation, it is recommended that the authors supplement several typical cases to illustrate the judgment differences among different annotators and the final adjudication process, thereby enhancing interpretability and transparency.
2. The current definition of the Bias-Index relies on a limited set of reference models, and directly subtracting the Bias-Index from the Macro-F1 score to obtain the "Overall" indicator may lead to issues of dimension inconsistency and sensitivity in multi-model scenarios. It is suggested that the authors verify the robustness of this indicator on a larger model set and supplement other fairness metrics.
3. The paper’s results show obvious variations in task difficulty across different domains, yet the authors do not explain whether task balancing or weighting was performed. When comparing across domains, comprehensive scores may be affected by the distribution of domain samples, thereby impacting overall fairness.

**Questions:**

1. You mentioned that the annotation process underwent multiple rounds of review and reported a relatively high consistency metric. When discrepancies arise among annotators, is there a fixed adjudication process or arbitration mechanism in place? If yes, could you supplement typical cases in the appendix to help readers understand how to handle criteria with strong subjectivity?
2. Is this metric stable when the number or type of reference models changes? Have sensitivity analyses been conducted, or has the consistency of calculating the bias-index in a larger model pool been tested?

---

> ### Author Response · Authors · 2025-11-21
> **Response to Reviewer pJFi (Part 1/3)**
>
> Thank you for the detailed review! We address the reviewer’s identified weaknesses/questions and will include these discussions into our manuscript.
>
> > The current definition of the Bias-Index relies on a limited set of reference models, and directly subtracting the Bias-Index from the Macro-F1 score to obtain the "Overall" indicator may lead to issues of dimension inconsistency and sensitivity in multi-model scenarios. It is suggested that the authors verify the robustness of this indicator on a larger model set and supplement other fairness metrics.
>
> >Is this metric stable when the number or type of reference models changes? Have sensitivity analyses been conducted, or has the consistency of calculating the bias-index in a larger model pool been tested?
>
> We appreciate the reviewer’s thoughtful and important suggestion on Bias-Index. We fully agree that the ideal way to stress-test any bias metric is to compute it over a large and diverse set of reference models. In practice, however, computing the Bias-Index (or any similar fairness metric that relies on human expert ground-truth) is inherently constrained by the availability of model-generated responses but also the expert-annotated judgements for all model generations.
>
> In ProfBench, expert annotators provided binary fulfillment judgments for responses generated by the three models that, at the time of data collection (July 2025), represented the strongest proprietary and open-weight systems on these professional tasks (OpenAI o3, xAI Grok4, and DeepSeek R1-0528). Obtaining comparable high-quality human expert judgments for many additional models would require re-running the full annotation pipeline on thousands of new generations, which is prohibitively expensive and time-consuming given the PhD/MBA-level expertise required.
>
> Nevertheless, we believe the current three-model setup provides a sufficiently meaningful and interpretable signal of bias for the following reasons:
> - The three reference systems come from three distinct providers/organizations with different training data, objectives, and output styles (OpenAI, xAI, DeepSeek), providing reasonable coverage of the proprietary vs. open-weight divide that matters most for fairness concerns in the community.
> - As shown in Table 2,  the top-performing judges (gpt-oss-120b and gemini-2.5-pro) exhibit max–min differences of less than 1%, suggesting no significant preference for any single provider.
>
> That said, in order to further address the reviewer’s concern on fairness diagnostics, we propose complementing Bias index with two additional metrics:
> - Standard deviation of biases: the standard deviation of biases collected across 3 models. This is less sensitive to an extreme outlier than max - min while penalizing systematic spread.
> - Provide-level fairness gap: We compute the gap between average bias toward the open-weight model (R1) versus the average bias toward the two proprietary models (o3, Grok4).
>
> | Model                      | Standard Deviation of Biases | Provider-level Fairness Gap |
> |:-----------------------------------:|:------------------------------:|:------------------------------:|
> | gpt-4.1                               | 0.37%                         | 0.65%                        |
> | gemini-2.5-flash                 | 0.12%                         | 0.25%                        |
> | gpt-5                                   | 0.51%                        | 0.30%                        |
> | o3                                       | 0.65%                        | 0.89%                        |
> | gpt-oss-120b                      | 0.22%                        | 0.15%                        |
> | deepseek-v3.1                    | 0.29%                        | 0.45%                        |
>
>
> The summary of results with the new metrics are shown above. The top judges (as identified in our original analysis) achieve consistently small biases in these new setups, with gpt-oss-120b achieving 0.22% in standard deviation of biases and 0.15% in fairness gap. We believe these results strengthen the fairness analysis while remaining fully grounded in the high-quality expert annotations. Again, we appreciate the reviewer’s suggestion and will include these new metrics and the corresponding discussion in the manuscript.

---

> ### Author Response · Authors · 2025-11-21
> **Response to Reviewer pJFi (Part 2/3)**
>
> > The paper’s results show obvious variations in task difficulty across different domains, yet the authors do not explain whether task balancing or weighting was performed. When comparing across domains, comprehensive scores may be affected by the distribution of domain samples, thereby impacting overall fairness.
>
> Thank you for this insightful comment. We agree that observed differences in average performance across the four domains raise a legitimate question about whether domain-wise dissection is a fair way to report the score.
>
> Our design philosophy for ProfBench was to faithfully reflect the authentic difficulty and nature of real-world professional work in each field, rather than to artificially equalize difficulty through task selection, rubric pruning, or post-hoc reweighting. Forced equalization would risk several undesirable side effects: (i) discarding the hardest (and often most diagnostically valuable) tasks in a domain, (ii) inflating scores in inherently more complex scientific domains by relaxing rubrics, or (iii) introducing arbitrary weighting schemes that lack objective grounding and reduce transparency.
>
> Henceforth we did not make any specific intervention on the collected data, except that we match the number of tasks in each domain to be equal. At the same time, we report the comprehensive domain-wise / reasoning type-wise performance of all models, given that the overall score -  like any other aggregation methods - inevitably distills useful information about subcategory-level performances. Our design principle directly aligns with subcategories in prior benchmarks that focus on professional tasks, such as conversation theme-wise analysis in HealthBench [1]  and machine-learning problem domains in PaperBench [2].
>
> That said, we have additionally examined whether the observed absolute difficulty gaps actually distort the ranking or relative capability assessment of models when scores are aggregated. Two sets of additional analyses (conducted post-submission) show that our aggregation scheme does not harm the validity of cross-model comparisons:
> For each domain, we normalize all reasoning model scores by dividing with the best model score in that domain (i.e., setting the best model score = 1.0). We then compute the deviation of these normalized scores separately for each domain. The resulting variances are consistently similar to each other across all 4 domains:
> | Category             | Physics | Chemistry | Finance | Consulting |
> |----------------------|---------|-----------|---------|------------|
> | Standard Deviation   | 0.11    | 0.10      | 0.11    | 0.09       |
>
> This indicates that, regardless of absolute difficulty, the performance hierarchy and gaps between strong and weak models remain stable across all four domains. In addition, we also analyze the rank correlation between the performance in each domain and the overall aggregated score:
>
> | Category                   | Physics | Chemistry | Finance | Consulting |
> |----------------------------|---------|-----------|---------|------------|
> | Spearman’s rank correlation | 0.85    | 0.79      | 0.91    | 0.89       |
>
> In all 4 dimensions, the correlation marks close to or beyond 0.8, indicating very strong correlation. This directly evidences that the overall aggregated score faithfully reflects model capabilities in all domains, rather than systematically penalizing a specific subset. Taken together, these results show that our domain-based categorization and aggregation is diagnostically valid. To more directly address the reviewer’s concern and promote flexible usage of ProfBench, we will incorporate the normalized variance and rank correlation analysis in our revised version of the paper.
>
> [1] Rahul K. Arora, Jason Wei, Rebecca Soskin Hicks, Preston Bowman, Joaquin Quiñonero-Candela, Foivos Tsimpourlas, Michael Sharman, Meghan Shah, Andrea Vallone, Alex Beutel, Johannes Heidecke, Karan Singhal (2025). HealthBench: Evaluating Large Language Models Towards Improved Human Health. https://arxiv.org/abs/2505.08775
>
> [2] Giulio Starace, Oliver Jaffe, Dane Sherburn, James Aung, Jun Shern Chan, Leon Maksin, Rachel Dias, Evan Mays, Benjamin Kinsella, Wyatt Thompson, Johannes Heidecke, Amelia Glaese, Tejal Patwardhan. (2025). PaperBench: Evaluating AI's Ability to Replicate AI Research. ICML 2025.

---

> ### Author Response · Authors · 2025-11-21
> **Response to Reviewer pJFi (Part 3/3)**
>
> > The paper describes the annotation process and consistency metrics in detail, but lacks qualitative demonstration of controversial samples and explanations of the adjudication mechanism. Given the subjectivity of rubric-based evaluation, it is recommended that the authors supplement several typical cases to illustrate the judgment differences among different annotators and the final adjudication process, thereby enhancing interpretability and transparency.
>
> > You mentioned that the annotation process underwent multiple rounds of review and reported a relatively high consistency metric. When discrepancies arise among annotators, is there a fixed adjudication process or arbitration mechanism in place? If yes, could you supplement typical cases in the appendix to help readers understand how to handle criteria with strong subjectivity?
>
> We generally ask annotators to phrase prompts and rubrics in ways to require objective answers (e.g. making sure calculations are correct in a multi-step chemical equation), in order to reduce situations where strong subjectivity influences the fair assessment of responses. This is supported by the Human Validation experiment in Section 4.1, where three annotators independently grade responses based on a set of rubrics and achieved Fleiss’ $\kappa$ = 0.912, suggesting that the rubrics can be applied objectively.
> The multiple rounds of review refer to the process in which the prompt and rubrics were iteratively improved. Below is the description accompanied by a running example from Physics PhD pool:
> 1. **Annotator A submits an initial prompt:** : Consider a five-level "quadpod" atom with four long-lived states $\lvert 1\rangle,\lvert 2\rangle,\lvert 3\rangle,\lvert 4\rangle$ and one excited state $\lvert e\rangle$. In the rotating-wave frame ($\hbar=1$), …
> 2. **Reviewer B recommends Major prompt edits**:
> ```
> The prompt is mathematically precise, defines all relevant symbols, and fixes a gauge, making the problem well-posed and non open-ended. It clearly states the task of computing the $3\times3$ non-Abelian Berry holonomy for a specified loop.
> For improved clarity and rubric alignment, consider:
> - Stating explicitly that the holonomy is to be reported in the ordered dark basis $(|d_1\rangle,|d_2\rangle,|d_3\rangle)$ evaluated at $(\alpha,\gamma)=(0,0)$.
> - Noting that G is interpreted modulo $2\pi$ due to the periodic nature of $\gamma$.
> - Indicating whether the expected answer should be a fully simplified closed-form matrix in terms of $\theta,\phi,\chi,G$, or if an ordered product form is acceptable.
> - Optionally mentioning that the resulting holonomy lies in $\mathrm{SU}(3)$ for this loop. …
> ```
> 3. **Annotator A makes changes based on Reviewer B’s feedback**: Thank you for the detailed feedback! I have incorporated all the changes suggested by the reviewer. I have also changed scaffoldings and split multistep tasks into separate tasks. Hopefully the quality of the prompt has been improved.
> 4. **Reviewer B approves prompt**
> 5. **Annotator A submits an initial rubrics**: [{"description":"States the explicit orthonormal degenerate dark-state basis \n\\[|d_1\\rangle =\\frac{ …}]
> 6. **Reviewer B recommends minor rubric edits:** Overall, the rubric covers the full logical progression from defining the dark-state basis and gauge potentials to computing the holonomy. However, several criteria reference variables, definitions, or results from earlier criteria without reintroducing them. Each criterion should be self-contained, including all necessary definitions and expressions, so the model does not need to look back. I included my revisions for each section and checked them as well.
> 7. **Annotator A makes changes to rubrics**: [{"description":"States the explicit orthonormal degenerate dark-state basis \n\\[|d_1\\rangle =\\frac{ …}]
> 8. **Reviewer B approves rubrics**
> 9. **Annotator A annotates 3 model responses based on rubrics**
> 10. **Reviewer B recommends minor changes to model response annotations**
> 11. **Annotator A updates annotation 3 model responses based on rubrics**
> 12. **Reviewer B approves response annotations**
>
> For a portion of the dataset, steps 9 to 12 were also done by two other annotators (Annotator C and D)  from the same pool (e.g. Physics PhD). Annotations by Annotator C and D are only used to estimate annotator agreement rates. For the dataset, we only use data from Annotator A, who developed the prompt, rubrics and annotated model response against the rubrics, as supported by Reviewer B. Note that Reviewer B is typically one reviewer but can at times be different reviewers at each stage if the original reviewer is unavailable since each task takes 10-20 hours across the three stages. Steps 3 to 4 / 7 to 8 / 11 to 12  can be repeated a few times until the reviewer is satisfied.
>
> > Final Comments
>
> We hope our response has addressed your initial concerns and we would appreciate if you could consider raising your appraisal in light of it.

---

> ### Comment · Reviewer_pJFi · 2025-11-26
>
> Thanks for your detailed response. I have raised my score accordingly.

---

### Official Review · Reviewer_knFu · 2025-10-29

**Soundness:** 3
**Presentation:** 3
**Contribution:** 3
**Rating:** 8
**Confidence:** 4

**Summary:**

The paper introduces a new benchmark (ProfBench) to test LLMs on PhD/MBA-level tasks that require specialized knowledge. Unlike prior benchmarks focusing on quickly verifiable problems, ProfBench focuses on open-ended tasks like writing reports that require expertise. The paper’s contributions include: (1) establishing a multi-domain, expert-annotated rubric benchmark; (2) assessing LLMs both as report generators and as judges; (3) and proposing techniques for LLM-based grading that aims to reduce evaluation cost and bias. The experiments demonstrate that LLM judges can grade responses with reasonable agreement to human experts.

**Strengths:**

- A major strength is the exploration of LLMs as automated judges. Building on work in rubric-evaluation, the authors propose a framework to have LLMs determine if a given response satisfies each expert criterion. The framework aims to reduce self-enhancement bias (i.e., where LLM judges would favor responses from the same model or provider), as well as API costs.
- ProfBench benchmarks LLMs as report generators in scenarios that mirror actual professional workflows, requiring multi-step reasoning and synthesizing information from multiple reference documents.
- Ablation experiments demonstrate the importance of reference documents for model performance.

**Weaknesses:**

- The benchmark’s scoring relies on LLM judges, and while the authors do measure agreement with human annotators, it’s shown that the best judge isn’t near perfect (<80% Macro-F1 overall). There’re risks that LLM-judges might miss nuanced criteria fulfillment or penalize creative answers. The paper doesn’t deeply discuss failure modes of the LLM-judge.
- ProfBench covers only four domains, leaving out several important domains of professional reasoning — notably, the legal, health, and engineering domains.
- Relatedly, the paper omits some relevant benchmarks involving professional tasks in the legal and engineering domains, which are not covered by the benchmark:

      Chilton, A., Guha, N., Nyarko, J., Ho, D., Ré, C., Narayana, A., ... & Peters, A. (2023). LegalBench: A Collaboratively Built Benchmark for Measuring Legal Reasoning in Large Language Models. Advances in Neural Information Processing Systems, 36.

      Zhou, X., Wang, X., He, Y., Wu, Y., Zou, R., Cheng, Y., ... & Zhao, J. (2025). EngiBench: A Benchmark for Evaluating Large Language Models on Engineering Problem Solving. arXiv preprint arXiv:2509.17677.

- The paper does not discuss any strategy for preventing test data leakage or overfitting. In contrast, some prior benchmarks like MMLU-Pro introduced hard, out-of-distribution questions to stay ahead of models.

- The paper’s evaluation doesn’t include an overall quality judgment beyond summing criteria. While rubric scoring is objective, it might not capture important aspects of professional work such (e.g., originality or creativity) that are hard to enumerate. This is a philosophical weakness of rubric-based evaluation in general, and acknowledging this potential limitation would improve the work.

**Questions:**

- Is there a risk of saturation with the best model scoring over 65%? Is the benchmark future-proof? Should there be a “hard” subset of prompts that are more adversarial in nature?

-  Could the authors clarify their rationale for selecting the four domains for ProfBench? For example, was the health or legal domains excluded due to difficulty in obtaining annotations or because existing benchmarks like HealthBench or LegalBench already cover this ground? If the goal is to provide a broad measure of professional knowledge, adding other professional domains seems valuable.

---

> ### Author Response · Authors · 2025-11-21
> **Response to Reviewer knFu (Part 1/4)**
>
> Thanks for recognizing our effort in improving evaluation of LLMs both as judges and report generators! We address the reviewer’s identified weaknesses/questions and will include these discussions into our manuscript.
>
> > The benchmark’s scoring relies on LLM judges, and while the authors do measure agreement with human annotators, it’s shown that the best judge isn’t near perfect (<80% Macro-F1 overall). There’re risks that LLM-judges might miss nuanced criteria fulfillment or penalize creative answers. The paper doesn’t deeply discuss failure modes of the LLM-judge.
>
> We agree that LLM judges are not perfect and may miss nuanced criteria fulfillment. However, we note that the best LLM-Judge in this paper (macro-F1 of 78.2) perform at a similar level compared to LLM Judges used in HealthBench [2] and PaperBench [2] which have macro-F1 at 70.9 and 83.0 respectively.
>
> To better understand where LLM Judges fall short, we include the category-specific F1 (Extraction, Reasoning and Style) in Table 2. Across these categories, models generally perform better on Extraction, followed by Reasoning and finally Style. This suggests that many LLM-Judges are weaker in judging whether responses fulfil stylistic requirements (e.g. writing an investment memo in a specific format; returning chemistry calculations to the requested precision) since they are sometimes more nuances and could involve checking different parts of the response (e.g. all concentration values to two decimal places) rather than Extraction tasks that are generally more focused on a specific segment within the response.
>
> In addition, we did a qualitative analysis of some reasoning traces, where the human annotators disagree with the optimal LLM judge (GPT-OSS-120B, high reasoning effort for Chemistry/Physics/Style, low otherwise).
>
> Here are some patterns we observed that explains the shortcomings of the LLM-Judge:
>
> 1. **LLM Judge sometimes does not recognize the equivalence of certain statements.**
>
> a. **Criterion**: Identifies the mechanism of the ylide formation as SN2 substitution.
>
> b. **Explanation**: In this case, the Bimolecular (i.e. two reactants) Nucleophilic substitution (SN2) reaction produces both the ylide and the phosphonium salt but the LLM judge wrongly focusses on only the phosphonium salt
>
> c. **Reasoning Trace**: The user asks: \"Evaluate whether the response above satisfies this criterion: Identifies the mechanism of the ylide formation as SN2 substitution.. Only answer Yes or No.\"\n\nWe need to examine the provided response (the synthesis plan) and see if it identifies the mechanism of the ylide formation as SN2 substitution.\n\nThe ylide formation step is the formation of the ylide from the phosphonium salt and n-BuLi. The question: \"Identifies the mechanism of the ylide formation as SN2 substitution.\" Actually, the ylide formation is deprotonation of the phosphonium salt by n-BuLi, which is not SN2 substitution but base deprotonation. However, the response includes a step \"Preparation of Isopropyltriphenylphosphonium Bromide\" which is an SN2 substitution (PPh3 attacks alkyl bromide). But the question specifically: \"Identifies the mechanism of the ylide formation as SN2 substitution.\" The ylide formation is the step where the phosphonium salt is deprotonated to form the ylide, which is not SN2. The response says:\n\n- Step 3: Preparation of Isopropyltriphenylphosphonium Bromide: \"Mechanism: SN2 attack by P on alkyl halide\". That's the formation of the phosphonium salt, not the ylide.\n\n- Step 4: Wittig Reaction to Form Dienophile: \"Mechanism: (a) Deprotonation by n-BuLi forms ylide. (b) Ylide attacks carbonyl...\". It does not state that the ylide formation is SN2 substitution; it says deprotonation.\n\nThus, the response does not identify the mechanism of ylide formation as SN2 substitution. It identifies the mechanism of phosphonium salt formation as SN2, not the ylide formation.\n\nThus, the answer to the criterion is No.\n\nThus output \"No\"."
>
> (Continued in next part of Response)

---

> ### Author Response · Authors · 2025-11-21
> **Response to Reviewer knFu (Part 2/4)**
>
> 2. **LLM judge sometimes mistakes implied statements for explicit statements.**
>
> a. **Criterion**: Defines the theoretical moles of hydrazine as\n\\[ n_{N_2H_4} = \\frac{n_{e^-}}{\\text{Number of electrons}} \\]
>
> b. **Explanation**: This was not explicitly mentioned in the response but utilized as an assumption in subsequent calculations.
>
> c. **Reasoning Trace**: We need to parse the question. The user gave a response that includes the calculation of theoretical production rate and defines theoretical moles of N2H4 as something. The criterion: \"Defines the theoretical moles of hydrazine as \\[ n_{N_2H_4} = \\frac{n_{e^-}}{\\text{Number of electrons}} \\]. Only answer Yes or No.\"\n\nWe need to evaluate whether the response above satisfies this criterion. The response includes a line: \"The theoretical production rate assumes 100% Faradaic efficiency. Total moles of electrons = Q / F = 0.23768 / 96485 = 2.4633 × 10⁻⁶ mol Theoretical moles of N₂H₄ = 2.4633 × 10⁻⁶ / 4 = 6.1583 × 10⁻⁷ mol\". This is indeed defining the theoretical moles of hydrazine as n(N2H4) = n_e / number of electrons (where number of electrons = 4). The response does define it in that way. The criterion is to define theoretical moles of hydrazine as n(N2H4) = n_e / Number of electrons. The response does exactly that: \"Theoretical moles of N₂H₄ = 2.4633 × 10⁻⁶ / 4 = 6.1583 × 10⁻⁷ mol\". It effectively defines it as n(N2H4) = n_e / 4. So the response satisfies the criterion. So answer \"Yes\".
>
> 3. **LLM judge sometimes fails to integrate information from different parts of the response.**
>
> a. **Criterion**: Proposes a substitution reaction with isopropanol and phosphorus tribromide in diethyl ether.
>
> b. **Explanation**: The response contains two parts - a. "All steps are designed to be carried out under standard laboratory conditions, with diethyl ether as the primary solvent" and b. “Preparation of isopropyl bromide (iPrBr): Mix isopropanol (27 g, 0.45 mol) with phosphorus tribromide (33 g, 0.12 mol) in a round-bottom flask." but the LLM judge missed the first part.
>
> c. **Reasoning Trace**: The user asks:\n\n\"Evaluate whether the response above satisfies this criterion: Proposes a substitution reaction with isopropanol and phosphorus tribromide in diethyl ether.. Only answer Yes or No.\"\n\nWe need to read the instruction: The user wants us to evaluate whether the response above satisfies this criterion: \"Proposes a substitution reaction with isopropanol and phosphorus tribromide in diethyl ether.\"\n\nWe have a response above (the assistant's previous answer) which includes a synthesis plan. The question is: does the response propose a substitution reaction with isopropanol and phosphorus tribromide in diethyl ether? Let's see.\n\nIn the response, step 1 is \"Preparation of isopropyl bromide (iPrBr): Mix isopropanol (27 g, 0.45 mol) with phosphorus tribromide (33 g, 0.12 mol) in a round-bottom flask. Heat gently (40-50\u00b0C) with stirring for 2 hours. Distill the product under reduced pressure to isolate iPrBr (~33 g expected after 60% yield).\"\n\nIt doesn't explicitly mention diethyl ether as the solvent for this step. It says \"in a round-bottom flask.\" It does not say diethyl ether. The question criterion: \"Proposes a substitution reaction with isopropanol and phosphorus tribromide in diethyl ether.\" The response proposes a substitution reaction between isopropanol and PBr3, but does it specify diethyl ether as the solvent? No. It just says \"in a round-bottom flask.\" It does not mention diethyl ether. So the criterion is not satisfied because the substitution reaction is not specifically described as being in diethyl ether. However, the question might be ambiguous: does the criterion require the reaction to be performed in diethyl ether? The response does propose the substitution reaction, but not in diethyl ether. So we answer No.\n\nThus answer: No."
>
> **Reflection**
>
> While these shortcomings highlight that LLM-Judges are not perfect oracles, the errors tend to be systematic rather than random and therefore may have less effect on relative comparisons between models as opposed to the absolute value. As we noted in lines 331 to 332, we only saw a 0.7 to 1.3% gap between judge-predicted and human-annotated performance across 3 models that were annotated by our LLM judge and professional human annotators.

---

> ### Author Response · Authors · 2025-11-21
> **Response to Reviewer knFu (Part 3/4)**
>
> > ProfBench covers only four domains, leaving out several important domains of professional reasoning — notably, the legal, health, and engineering domains.
>
> > Relatedly, the paper omits some relevant benchmarks involving professional tasks in the legal and engineering domains, which are not covered by the benchmark:
>
> > Could the authors clarify their rationale for selecting the four domains for ProfBench? For example, was the health or legal domains excluded due to difficulty in obtaining annotations or because existing benchmarks like HealthBench or LegalBench already cover this ground? If the goal is to provide a broad measure of professional knowledge, adding other professional domains seems valuable.
>
> Thank you for pointing us to the LegalBench and EngiBench works - we will cite them in our paper.
> We select the domains of Chemistry Research PhD, Physics Research PhD, Consulting MBA and Finance MBA based on three factors: a. Whether other works have covered such domains previously, b. Real-world usefulness of these domains and c. Whether we could recruit suitable annotators to robustly annotate for these fields.
>
> For the first factor, we focussed on domains that have not been previously worked on by earlier works [1], [2], [3], [4] in order to show fresh capabilities in new settings, which we believe to be more helpful to the community.
>
> For the second factor, we wanted to measure LLM capabilities that real-world users care about. For instance, many existing advanced Chemistry and Physics evaluations (e.g. GPQA [5] and HLE [6]) mostly measure exam-style problems that can be answered in a Multiple-choice or a short span format. For many real-world physical scientists, answering these exam questions are less helpful in their work compared to assisting them in their research planning, as covered in ProfBench. In addition, we want to focus on some enterprise use-cases and we found that capabilities relevant to Consulting MBA and Finance MBA are relevant to many white-collar professions - meaning that measuring LLM capabilities in these domains can proxy their usefulness for many real-world professional users of AI.
>
> For the third factor, we were constrained to annotators that we could feasibly recruit for to do these tasks within our research timeframe and budget. Given these constraints, we opted to focus on the set of 4 domains, each with a useful number of tasks (20 tasks / domain, each taking 10-20 hours of expert annotations) rather than covering a large set of domains but with a small number of tasks in each (e.g. DeepResearch-Bench [7]  covers 22 domains but only has an average of 2.3 tasks in each domain in English). We do not have near-term plans to expand the number of domains for newly-annotated tasks although it is possible in the medium or long term. In the short term, we recommend researchers to utilize ProfBench along other benchmarks [1], [2], [3], [4] where appropriate for their research goals.
>
> [1] Rahul K. Arora, Jason Wei, Rebecca Soskin Hicks, Preston Bowman, Joaquin Quiñonero-Candela, Foivos Tsimpourlas, Michael Sharman, Meghan Shah, Andrea Vallone, Alex Beutel, Johannes Heidecke, Karan Singhal (2025). HealthBench: Evaluating Large Language Models Towards Improved Human Health. https://arxiv.org/abs/2505.08775
>
> [2] Giulio Starace, Oliver Jaffe, Dane Sherburn, James Aung, Jun Shern Chan, Leon Maksin, Rachel Dias, Evan Mays, Benjamin Kinsella, Wyatt Thompson, Johannes Heidecke, Amelia Glaese, Tejal Patwardhan. (2025). PaperBench: Evaluating AI's Ability to Replicate AI Research. ICML 2025.
>
> [3] Chilton, A., Guha, N., Nyarko, J., Ho, D., Ré, C., Narayana, A., ... & Peters, A. (2023). LegalBench: A Collaboratively Built Benchmark for Measuring Legal Reasoning in Large Language Models. NeurIPS 2023.
>
> [4] Zhou, X., Wang, X., He, Y., Wu, Y., Zou, R., Cheng, Y., ... & Zhao, J. (2025). EngiBench: A Benchmark for Evaluating Large Language Models on Engineering Problem Solving. https://arxiv.org/abs/2509.17677
>
> [5] David Rein, Betty Li Hou, Asa Cooper Stickland, Jackson Petty, Richard Yuanzhe Pang, Julien Dirani, Julian Michael, Samuel R. Bowman. (2024) GPQA: A Graduate-Level Google-Proof Q&A Benchmark. COLM 2024.
>
> [6] Long Phan, Alice Gatti, Ziwen Han, Nathaniel Li … Summer Yue, Alexandr Wang,  Dan Hendrycks. 2025. Humanity's Last Exam. https://arxiv.org/abs/2501.14249
>
> [7] Mingxuan Du, Benfeng Xu, Chiwei Zhu, Xiaorui Wang, and Zhendong Mao. Deepresearch bench: A
> comprehensive benchmark for deep research agents, 2025. URL https://arxiv.org/abs/2506.11763.

---

> ### Author Response · Authors · 2025-11-21
> **Response to Reviewer knFu (Part 4/4)**
>
> > The paper does not discuss any strategy for preventing test data leakage or overfitting. In contrast, some prior benchmarks like MMLU-Pro introduced hard, out-of-distribution questions to stay ahead of models.
>
> We clarify that all prompts and rubrics were newly annotated, so that they introduce unseen challenges to LLMs, meaning that it is substantially more difficult compared to MMLU-Pro, which has the top model scoring 93.2% while the top model on ProfBench only scores 65.9%. We share your concern about test data leakage/overfitting and to mitigate this possibility, we reserve half of the dataset as the private test set as described in lines 228-229.
>
> > The paper’s evaluation doesn’t include an overall quality judgment beyond summing criteria. While rubric scoring is objective, it might not capture important aspects of professional work such (e.g., originality or creativity) that are hard to enumerate. This is a philosophical weakness of rubric-based evaluation in general, and acknowledging this potential limitation would improve the work.
>
> We believe that aspects such as originality and creativity are hard to objectively measure (even for humans, they tend to be very subjective). We specifically ask annotators to phrase prompts in ways that do not necessarily require creative answers (e.g. making sure calculations are correct in a multi-step chemical equation). As models gradually become performant in objective tasks in ProfBench, we agree that the next step of evaluation might involve measuring creativity of such professional tasks (e.g. finding creative chemical experimental approaches or structuring novel financial deals), which can build upon ProfBench and prior works on measuring creativity [1].
>
> [1] Ximing Lu, Melanie Sclar, Skyler Hallinan, Niloofar Mireshghallah, Jiacheng Liu, Seungju Han, Allyson Ettinger, Liwei Jiang, Khyathi Chandu, Nouha Dziri, Yejin Choi. 2025, AI as Humanity's Salieri: Quantifying Linguistic Creativity of Language Models via Systematic Attribution of Machine Text against Web Text. ICLR 2025.
>
> > Is there a risk of saturation with the best model scoring over 65%? Is the benchmark future-proof? Should there be a “hard” subset of prompts that are more adversarial in nature?
>
> We believe that the current difficulty (50% +/- 15% for most models) is useful for many model trainers to get effective signals and improve. To reach this target difficulty, we focus on the **easiest** setup for report generation (Section 5), where models are given grounding documents that were truncated by human experts (to max. 20 pages each) to make it easy for LLMs to locate where the relevant information is.  As models get better, we can adapt the benchmark to make it more challenging and avoid saturation. One approach is to use the original documents (up to 800+ pages long each) to better test their long context processing capabilities as many models get more than 128k context window size.
>
> Furthermore, we can even remove the grounding documents altogether and force models to either search for relevant documents or answer prompts through parametric knowledge alone. An ablation study that we conducted in Section 6 shows that such settings cause model performance to drop by 4.9% to 11.9%. This means that once the current version of ProfBench becomes too easy for the strongest LLMs, we can easily flip a switch to make ProfBench substantially more challenging and future-proof. For instance, the top performing LLM only reaches 52% when forced to rely on parametric knowledge.

---

> > ### Comment · Reviewer_knFu · 2025-11-26
> >
> > The authors' thorough reply adequately addresses most of my concerns. They provide an analysis of the LLM-Judge's reliability, justify their chosen domain coverage, and agreed to include relevant missing benchmark citations (LegalBench and EngiBench). Their discussion of steps to avoid overfitting and future-proofing the benchmark is reassuring. However, the limitation of rubric-based evaluation vs. a holistic quality assessment remains partly resolved. I'm satisfied with the responses and will maintain the score.

---

### Official Review · Reviewer_yfAJ · 2025-10-31

**Soundness:** 4
**Presentation:** 3
**Contribution:** 3
**Rating:** 8
**Confidence:** 3

**Summary:**

The paper proposes a benchmark including response-criterion pairs to evaluate professional knowledge across multiple fields. It introduces an efficient LLM-as-judge evaluation framework that mitigates self-enhancement bias. The authors evaluate current LLM performance on both criterion fulfillment classification and response generation for these challenging tasks.

**Strengths:**

1. The benchmark covers multiple scientific domains and evaluates knowledge storage and complex reasoning capabilities. Its expert-designed criteria facilitate precise, granular assessment of LLM performance on challenging tasks.
2. The paper assesses a wide range of LLMs to provide comprehensive performance benchmarks. The experimental design encompasses comparisons across model accessibility (open-source and closed-source), scale, and reasoning capabilities.
3. The high-quality annotators group and reliable rubric creation pipeline guarantee the dataset quality.

**Weaknesses:**

1. In Section 4, the LLM-as-judge is used as a binary classifier, with performance evaluated by F1 score. The target LLM is used to identify whether the provided criterion fulfills all the requirements to check the quality of the response. However, for such complex tasks, the F1 score only captures misalignment between the LLM and human experts. It does not reveal the LLM's internal understanding of the task or identify specific weaknesses.
2. In the rubric creation process from Section 3, the criteria creation and review stages are not described in detail. Both stages appear heavily dependent on annotator judgment, and it remains unclear how each proposed criterion contributes to the granular assessment of response quality.

**Questions:**

1. While using LLM to judge the criterion-fulfillment, is there any way to extract more information from the LLM performance for further failure mode analysis? Reasoning models are allowed to generate some inference steps prior to binary predictions. How might these reasoning traces be utilized to analyze misalignment between LLM predictions and human annotations?

---

> ### Author Response · Authors · 2025-11-21
> **Response to Reviewer yfAJ (Part 1/3)**
>
> Thank you for recognizing the care we put into constructing ProfBench! Below, we address the reviewer’s specific concerns and will include these discussions into our manuscript.
>
> > In the rubric creation process from Section 3, the criteria creation and review stages are not described in detail. Both stages appear heavily dependent on annotator judgment, and it remains unclear how each proposed criterion contributes to the granular assessment of response quality.
>
> Both stages do indeed depend heavily on annotator judgement, given the professional nature of the tasks, meaning that specific expertise (at an MBA/PhD level)  is needed to assess the suitability of the criteria. Therefore, we relied on our annotators to ensure that the rubrics capture all important aspects of a good response in a granular fashion, with further checks by one or more reviewers, who can pinpoint specific areas the original annotators can improve upon. Details of the annotation and review process can be found in Section 3 and Appendix C.
>
> The multiple rounds of review refers to the process in which the prompt and rubrics were iteratively improved. Below is the detailed description accompanied by a running example from Physics PhD pool::
>
> 1. **Annotator A submits an initial prompt:** : Consider a five-level "quadpod" atom with four long-lived states $\lvert 1\rangle,\lvert 2\rangle,\lvert 3\rangle,\lvert 4\rangle$ and one excited state $\lvert e\rangle$. In the rotating-wave frame ($\hbar=1$), …
> 2. **Reviewer B recommends Major prompt edits**:
> ```
> The prompt is mathematically precise, defines all relevant symbols, and fixes a gauge, making the problem well-posed and non open-ended. It clearly states the task of computing the $3\times3$ non-Abelian Berry holonomy for a specified loop.
> For improved clarity and rubric alignment, consider:
> - Stating explicitly that the holonomy is to be reported in the ordered dark basis $(|d_1\rangle,|d_2\rangle,|d_3\rangle)$ evaluated at $(\alpha,\gamma)=(0,0)$.
> - Noting that G is interpreted modulo $2\pi$ due to the periodic nature of $\gamma$.
> - Indicating whether the expected answer should be a fully simplified closed-form matrix in terms of $\theta,\phi,\chi,G$, or if an ordered product form is acceptable.
> - Optionally mentioning that the resulting holonomy lies in $\mathrm{SU}(3)$ for this loop. …
> ```
> 3. **Annotator A makes changes based on Reviewer B’s feedback**: Thank you for the detailed feedback! I have incorporated all the changes suggested by the reviewer. I have also changed scaffoldings and split multistep tasks into separate tasks. Hopefully the quality of the prompt has been improved.
> 4. **Reviewer B approves prompt:**
> 5. **Annotator A submits an initial rubrics**: [{"description":"States the explicit orthonormal degenerate dark-state basis \n\\[|d_1\\rangle =\\frac{ …}]
> 6. **Reviewer B recommends minor rubric edits:** Overall, the rubric covers the full logical progression from defining the dark-state basis and gauge potentials to computing the holonomy. However, several criteria reference variables, definitions, or results from earlier criteria without reintroducing them. Each criterion should be self-contained, including all necessary definitions and expressions, so the model does not need to look back. I included my revisions for each section and checked them as well.
> 7. **Annotator A makes changes to rubrics**: [{"description":"States the explicit orthonormal degenerate dark-state basis \n\\[|d_1\\rangle =\\frac{ …}]
> 8. **Reviewer B approves rubrics**
> 9. **Annotator A annotates 3 model responses based on rubrics**
> 10. **Reviewer B recommends minor changes to model response annotations**
> 11. **Annotator A updates annotation 3 model responses based on rubrics**
> 12. **Reviewer B approves response annotations**
>
> Annotator A is a single annotator who developed the prompt, rubrics and annotated model responses against the rubrics, as supported by Reviewer B. Note that Reviewer B is typically one reviewer but can at times be different reviewers at each stage if the original reviewer is unavailable since each task takes 10-20 hours across the three stages. Steps 3 to 4 / 7 to 8 / 11 to 12  can be repeated a few times until the reviewer is satisfied.
>
> During the rubric review process, both the annotator and reviewer are instructed to check that the rubrics fulfil a number of requirements (e.g. each can be graded in a binary fashion, should not overlap with other criteria and must collectively cover all important aspects of a good response). There is a longer list of requirements that we initially planned to include into the Appendix. However, some of this information constitutes confidential business-sensitive information for our data vendor and hence we were unable to include it into the submission.

---

> ### Author Response · Authors · 2025-11-21
> **Response to Reviewer yfAJ (Part 2/3)**
>
> > In Section 4, the LLM-as-judge is used as a binary classifier, with performance evaluated by F1 score. The target LLM is used to identify whether the provided criterion fulfills all the requirements to check the quality of the response. However, for such complex tasks, the F1 score only captures misalignment between the LLM and human experts. It does not reveal the LLM's internal understanding of the task or identify specific weaknesses.
>
> > While using LLM to judge the criterion-fulfillment, is there any way to extract more information from the LLM performance for further failure mode analysis? Reasoning models are allowed to generate some inference steps prior to binary predictions. How might these reasoning traces be utilized to analyze misalignment between LLM predictions and human annotations?
>
> We agree that the overall F1 score only captures misalignment between the LLM and human experts. In addition to this, we also include the category-specific F1 (Extraction, Reasoning and Style) in Table 2. Across these categories, models generally perform better on Extraction, followed by Reasoning and finally Style. This suggests that many LLM-Judges are weaker in judging whether responses fulfil stylistic requirements (e.g. writing an investment memo in a specific format; returning chemistry calculations to the requested precision) since they are sometimes more nuances and could involve checking different parts of the response (e.g. all concentration values to two decimal places) rather than Extraction tasks that are generally more focused on a specific segment within the response.
>
> Thank you for further recommending this analysis of reasoning traces! We did a qualitative analysis of some reasoning traces, where the human annotators disagree with the optimal LLM judge (GPT-OSS-120B, high reasoning effort for Chemistry/Physics/Style, low otherwise).
>
> Here are some patterns we observed that explains the shortcomings of the LLM-Judge:
>
> 1. **LLM Judge sometimes does not recognize the equivalence of certain statements.**
>
> a. **Criterion**: Identifies the mechanism of the ylide formation as SN2 substitution.
>
> b. **Explanation**: In this case, the Bimolecular (i.e. two reactants) Nucleophilic substitution (SN2) reaction produces both the ylide and the phosphonium salt but the LLM judge wrongly focusses on only the phosphonium salt
>
> c. **Reasoning Trace**: The user asks: \"Evaluate whether the response above satisfies this criterion: Identifies the mechanism of the ylide formation as SN2 substitution.. Only answer Yes or No.\"\n\nWe need to examine the provided response (the synthesis plan) and see if it identifies the mechanism of the ylide formation as SN2 substitution.\n\nThe ylide formation step is the formation of the ylide from the phosphonium salt and n-BuLi. The question: \"Identifies the mechanism of the ylide formation as SN2 substitution.\" Actually, the ylide formation is deprotonation of the phosphonium salt by n-BuLi, which is not SN2 substitution but base deprotonation. However, the response includes a step \"Preparation of Isopropyltriphenylphosphonium Bromide\" which is an SN2 substitution (PPh3 attacks alkyl bromide). But the question specifically: \"Identifies the mechanism of the ylide formation as SN2 substitution.\" The ylide formation is the step where the phosphonium salt is deprotonated to form the ylide, which is not SN2. The response says:\n\n- Step 3: Preparation of Isopropyltriphenylphosphonium Bromide: \"Mechanism: SN2 attack by P on alkyl halide\". That's the formation of the phosphonium salt, not the ylide.\n\n- Step 4: Wittig Reaction to Form Dienophile: \"Mechanism: (a) Deprotonation by n-BuLi forms ylide. (b) Ylide attacks carbonyl...\". It does not state that the ylide formation is SN2 substitution; it says deprotonation.\n\nThus, the response does not identify the mechanism of ylide formation as SN2 substitution. It identifies the mechanism of phosphonium salt formation as SN2, not the ylide formation.\n\nThus, the answer to the criterion is No.\n\nThus output \"No\"."
>
> (Continued in next part of Response)

---

> ### Author Response · Authors · 2025-11-21
> **Response to Reviewer yfAJ (Part 3/3)**
>
> 2. **LLM judge sometimes mistakes implied statements for explicit statements.**
>
> a. **Criterion**: Defines the theoretical moles of hydrazine as\n\\[ n_{N_2H_4} = \\frac{n_{e^-}}{\\text{Number of electrons}} \\]
>
> b. **Explanation**: this was not explicitly mentioned in the response but utilized as an assumption in subsequent calculations
>
> c. **Reasoning Trace**: We need to parse the question. The user gave a response that includes the calculation of theoretical production rate and defines theoretical moles of N2H4 as something. The criterion: \"Defines the theoretical moles of hydrazine as \\[ n_{N_2H_4} = \\frac{n_{e^-}}{\\text{Number of electrons}} \\]. Only answer Yes or No.\"\n\nWe need to evaluate whether the response above satisfies this criterion. The response includes a line: \"The theoretical production rate assumes 100% Faradaic efficiency. Total moles of electrons = Q / F = 0.23768 / 96485 = 2.4633 × 10⁻⁶ mol Theoretical moles of N₂H₄ = 2.4633 × 10⁻⁶ / 4 = 6.1583 × 10⁻⁷ mol\". This is indeed defining the theoretical moles of hydrazine as n(N2H4) = n_e / number of electrons (where number of electrons = 4). The response does define it in that way. The criterion is to define theoretical moles of hydrazine as n(N2H4) = n_e / Number of electrons. The response does exactly that: \"Theoretical moles of N₂H₄ = 2.4633 × 10⁻⁶ / 4 = 6.1583 × 10⁻⁷ mol\". It effectively defines it as n(N2H4) = n_e / 4. So the response satisfies the criterion. So answer \"Yes\".
>
> But let's examine more carefully: The criterion likely expects that the response includes a definition that the theoretical moles of hydrazine equals the moles of electrons divided by the number of electrons required per hydrazine molecule. The response includes "The charge passed corresponds to the theoretical amount of N₂H₄ that could be produced: Q = n·F·moles of N₂H₄" and then "Theoretical N₂H₄ = Q/(n·F)". This is equivalent to n_{N2H4} = (Q/F)/n = n_{e-}/n. So yes, they defined it.
>
> Thus answer: Yes.
>
> 3. **LLM judge sometimes fail to integrate information from different parts of the response.**
>
> a. **Criterion**: Proposes a substitution reaction with isopropanol and phosphorus tribromide in diethyl ether.
>
> b. **Explanation**: The response contains two parts - a. "All steps are designed to be carried out under standard laboratory conditions, with diethyl ether as the primary solvent" and b. “Preparation of isopropyl bromide (iPrBr): Mix isopropanol (27 g, 0.45 mol) with phosphorus tribromide (33 g, 0.12 mol) in a round-bottom flask." but the LLM judge missed the first part..
>
> c. **Reasoning Trace**: The user asks:\n\n\"Evaluate whether the response above satisfies this criterion: Proposes a substitution reaction with isopropanol and phosphorus tribromide in diethyl ether.. Only answer Yes or No.\"\n\nWe need to read the instruction: The user wants us to evaluate whether the response above satisfies this criterion: \"Proposes a substitution reaction with isopropanol and phosphorus tribromide in diethyl ether.\"\n\nWe have a response above (the assistant's previous answer) which includes a synthesis plan. The question is: does the response propose a substitution reaction with isopropanol and phosphorus tribromide in diethyl ether? Let's see.\n\nIn the response, step 1 is \"Preparation of isopropyl bromide (iPrBr): Mix isopropanol (27 g, 0.45 mol) with phosphorus tribromide (33 g, 0.12 mol) in a round-bottom flask. Heat gently (40-50\u00b0C) with stirring for 2 hours. Distill the product under reduced pressure to isolate iPrBr (~33 g expected after 60% yield).\"\n\nIt doesn't explicitly mention diethyl ether as the solvent for this step. It says \"in a round-bottom flask.\" It does not say diethyl ether. The question criterion: \"Proposes a substitution reaction with isopropanol and phosphorus tribromide in diethyl ether.\" The response proposes a substitution reaction between isopropanol and PBr3, but does it specify diethyl ether as the solvent? No. It just says \"in a round-bottom flask.\" It does not mention diethyl ether. So the criterion is not satisfied because the substitution reaction is not specifically described as being in diethyl ether. However, the question might be ambiguous: does the criterion require the reaction to be performed in diethyl ether? The response does propose the substitution reaction, but not in diethyl ether. So we answer No.\n\nThus answer: No."
>
> **Reflection**
>
> While these shortcomings highlight that LLM-Judges are not perfect oracles, the errors tend to be systematic rather than random and therefore may have less effect on relative comparisons between models as opposed to the absolute value. As we noted in lines 331 to 332, we only saw a 0.7 to 1.3% gap between judge-predicted and human-annotated performance across 3 models that were annotated by our LLM judge and professional human annotators.

---

### Official Review · Reviewer_2ui6 · 2025-11-01

**Soundness:** 3
**Presentation:** 3
**Contribution:** 3
**Rating:** 4
**Confidence:** 4

**Summary:**

The paper introduces ProfBench, a rubric-guided benchmark of 80 expert-curated tasks spanning four domains, Chemistry, Physics, Finance, and Consulting, with more than 7000 human-written response-criterion pairs. Domain experts create the tasks as well as weighted task-specific rubrics and label three “seed” model responses (o3, Grok4, R1-0528) per criterion as Yes/No, forming the ground truth for calibrating an LLM judge. The quality of the judge is measured by a metric that takes into account also the bias towards the same model family. Each task has the structure of report-writing with grounding documents, and removing the documents leads to worsening performance. The best model reaches an average score of 65.9%, with lowest performance on Physics (49.3%) and best performance on Consulting (80%).

**Strengths:**

- The tasks are realistic, complex, with grounding documents and created by domain experts, with reviewer feedback.
- The LLM judges are evaluated in a clear way that takes also into account the bias towards the same model family.
- Separate re-annotations show high inter-annotator agreement. Moreover, the LLM judge is highly reliable, with a tiny difference with human-annotated scores.

**Weaknesses:**

- GPT-5 already achieves a high score on Consulting (~80%), while Physics lags at ~49%. This suggests that one domain might saturate sooner than the other.
- The set-up is text-only, even if tool use might be helpful for some tasks, e.g. calculators, spreadsheets, code etc.
- Despite current difficulty, the text-only format may offer limited room for improvement as models become more capable.
- Domain coverage is narrow, covering only two science and two business domains.

**Questions:**

- What was the rationale for choosing the four domains? Do you plan to add others?
- Are you planning on performing evaluations with models that can use tools that are relevant for the benchmark tasks?
- The LLM judge evaluation depends on three seed models. I wonder whether the judge evaluation can change when picking different models. Do you have any thoughts on that?

---

> ### Author Response · Authors · 2025-11-21
> **Response to Reviewer 2ui6 (Part 1/3)**
>
> Thank you for the constructive feedback! Below, we address the mentioned feedback and will include these discussions into our manuscript.
>
> >  GPT-5  already achieves a high score on Consulting (~80%), while Physics lags at ~49%. This suggests that one domain might saturate sooner than the other.
>
> We believe that introducing a benchmark with uneven performance across novel domains (e.g. Consulting-related tasks or Physics-research tasks) can lead to insights for subsequent research, especially when we were not previously able to measure LLM performance on such domains. For instance, this can help researchers to focus efforts on some domains relative to others (e.g. more on Physics research). It can also incentivize researchers to come up with more difficult tasks in domains like Consulting since this work shows the community that SOTA LLMs can confidently do “analyst”-level work. For example, subsequent research can move up the value chain to measure LLM capabilities at associate-level, senior-level or even partner-level tasks. In this sense, showing the difference in performance of LLM across domains is a strength rather than a weakness of our work.
>
> > The set-up is text-only, even if tool use might be helpful for some tasks, e.g. calculators, spreadsheets, code etc.
>
> > Are you planning on performing evaluations with models that can use tools that are relevant for the benchmark tasks?
>
> We clarify that our experiments do include tool-use. The main tool we focus on in Section 5 is the document-ingestion tool, which allows LLMs to utilize information from PDF grounding documents that human expert annotations identify. We also conducted smaller-scale experiments with other tools such as the search tool in Section 6. We decided to de-prioritize further explorations on the search tool because the implementations of LLM search tool use are currently non-standardized. For instance, proprietary providers (e.g. OpenAI, Google and Anthropic) typically have their first-party search tool while open-weight models have to depend on third-party search tools (e.g. Exa AI). Such differences mean that when a model using search-tool is measured, it becomes challenging to understand whether model performance can be accounted for by search-tool capabilities, model capabilities or the integration of search-tools with models. In addition, we also considered and made initial attempts with other tools (e.g. code interpreter environment - which integrates spreadsheet tool and calculator tool), since we initially had a small portion of documents in CSV format. However, at the time of our experiments, only OpenAI had a code interpreter tool integrated into a publicly accessible API. Using such tools makes it challenging to fairly compare these models with a vast number of other models that do not use these tools. As the ecosystem for various tools matures, we certainly plan to incorporate further evaluations using such tools.
>
> > Despite current difficulty, the text-only format may offer limited room for improvement as models become more capable.
>
> We purposefully designed ProfBench to be text-only (i.e. no image or audio) in order to support recent popular open-weight models since many of them are text-only models (e.g. DeepSeek V3.1 / Kimi K2 / GPT-OSS). This ensures the open-source community can readily build upon these models and utilize ProfBench to measure and improve model capabilities relevant to LLM-judge and long report generation. As the open-weight model ecosystem develops with more models supporting multi-modal capabilities, followup works can build upon the foundations laid by ProfBench.

---

> ### Author Response · Authors · 2025-11-21
> **Response to Reviewer 2ui6 (Part 2/3)**
>
> > Domain coverage is narrow, covering only two science and two business domains.
>
> > What was the rationale for choosing the four domains? Do you plan to add others?
>
> We select the domains of Chemistry Research PhD, Physics Research PhD, Consulting MBA and Finance MBA based on three factors: a. Whether other works have covered such domains previously, b. Real-world usefulness of these domains and c. Whether we could recruit suitable annotators to robustly annotate for these fields.
>
> For the first factor, we focused on domains that have not been previously worked on by earlier works [1], [2], [3], [4] in order to show fresh capabilities in new settings, which we believe to be more helpful to the community.
>
> For the second factor, we wanted to measure LLM capabilities that real-world users care about. For instance, many existing advanced Chemistry and Physics evaluations (e.g. GPQA [5] and HLE [6]) mostly measure exam-style problems that can be answered in a Multiple-choice or a short span format. For many real-world physical scientists, answering these exam questions is less helpful in their work compared to assisting them in their research planning, as covered in ProfBench. In addition, we want to focus on some enterprise use-cases and we found that capabilities relevant to Consulting MBA and Finance MBA are relevant to many white-collar professions - meaning that measuring LLM capabilities in these domains can proxy their usefulness for many real-world professional users of AI.
>
> For the third factor, we were constrained to annotators that we could feasibly recruit for to do these tasks within our research timeframe and budget. Given these constraints, we opted to focus on the set of 4 domains, each with a useful number of tasks (20 tasks / domain, each taking 10-20 hours of expert annotations) rather than covering a large set of domains but with a small number of tasks in each (e.g. DeepResearch-Bench [7] covers 22 domains but only has an average of 2.3 tasks in each domain in English). We do not have near-term plans to expand the number of domains for newly-annotated tasks although it is possible in the medium or long term. In the short term, we recommend researchers to utilize ProfBench along other benchmarks [1], [2], [3], [4] where appropriate for their research goals.
>
>
> [1] Rahul K. Arora, Jason Wei, Rebecca Soskin Hicks, Preston Bowman, Joaquin Quiñonero-Candela, Foivos Tsimpourlas, Michael Sharman, Meghan Shah, Andrea Vallone, Alex Beutel, Johannes Heidecke, Karan Singhal (2025). HealthBench: Evaluating Large Language Models Towards Improved Human Health. https://arxiv.org/abs/2505.08775
>
> [2] Giulio Starace, Oliver Jaffe, Dane Sherburn, James Aung, Jun Shern Chan, Leon Maksin, Rachel Dias, Evan Mays, Benjamin Kinsella, Wyatt Thompson, Johannes Heidecke, Amelia Glaese, Tejal Patwardhan. (2025). PaperBench: Evaluating AI's Ability to Replicate AI Research. ICML 2025.
>
> [3] Chilton, A., Guha, N., Nyarko, J., Ho, D., Ré, C., Narayana, A., ... & Peters, A. (2023). LegalBench: A Collaboratively Built Benchmark for Measuring Legal Reasoning in Large Language Models. NeurIPS 2023.
>
> [4] Zhou, X., Wang, X., He, Y., Wu, Y., Zou, R., Cheng, Y., ... & Zhao, J. (2025). EngiBench: A Benchmark for Evaluating Large Language Models on Engineering Problem Solving. https://arxiv.org/abs/2509.17677
>
> [5] David Rein, Betty Li Hou, Asa Cooper Stickland, Jackson Petty, Richard Yuanzhe Pang, Julien Dirani, Julian Michael, Samuel R. Bowman. (2024) GPQA: A Graduate-Level Google-Proof Q&A Benchmark. COLM 2024.
>
> [6] Long Phan, Alice Gatti, Ziwen Han, Nathaniel Li … Summer Yue, Alexandr Wang,  Dan Hendrycks. 2025. Humanity's Last Exam. https://arxiv.org/abs/2501.14249
>
> [7] Mingxuan Du, Benfeng Xu, Chiwei Zhu, Xiaorui Wang, and Zhendong Mao. Deepresearch bench: A
> comprehensive benchmark for deep research agents, 2025. URL https://arxiv.org/abs/2506.11763.

---

> ### Author Response · Authors · 2025-11-21
> **Response to Reviewer 2ui6 (Part 3/3)**
>
> > The LLM judge evaluation depends on three seed models. I wonder whether the judge evaluation can change when picking different models. Do you have any thoughts on that?
>
> Thank you for the thoughtful question. We selected the three representative models (o3, Grok4 and R1-0528) intentionally as they represent the top-performing systems at the time of data collection (July 2025), spanning diverse providers and paradigms: two proprietary (o3 from OpenAI and Grok4 from xAI) and one open-weight (R1-0528 from DeepSeek). Ideally, sensitivity analysis would involve re-annotating responses from alternative reference sets with human experts, but this is resource-intensive due to the need for PhD/MBA-level judgments on thousands of new response-criterion pairs. Nevertheless, our existing analyses (including post-submission checks) demonstrate that the evaluations are robust and unlikely to change materially with different references:
>
> - As detailed in Section 4.2 and Table 2, top LLM-Judges (e.g., Gemini-2.5-Pro at 78.2%, gpt-oss-120b at 78.2%) exhibit extremely low bias-index values (less than 0.5% across these three reference models), meaning they align closely with human annotations regardless of the response's origin. This low spread suggests that good judges are impartial to the specific mix of proprietary/open-weight styles in our set. The impartiality of the LLM judges in ProfBench is in a large part contributed by the nature of the evaluation to decompose the LLM judgement into 15-60 rubric criteria per prompt that we independently score, then weighted together. Our approach results in substantially lower bias (in both relative terms <0.5% and absolute terms across 3 models <1.0% for our official gpt-oss-120B judge) compared to alternative approaches of applying LLM judges (e.g. picking preference among two responses [1,2,3]  or giving a single scalar score for each response [3]) where biases can be as high as 30 to 75%.
>
> - To directly test the null hypothesis that judge evaluations do not depend on the specific reference model identities, we conduct a permutation test (10,000 iterations) on judge evaluation. Specifically, we fix the judge predictions and human labels, randomly shuffle the model labels assigned to responses, and recompute the Bias Index under each shuffled labeling. For our final judge model, gpt-oss-120b, the observed Bias Index falls well within the null distribution, with two-sided p-value > 0.56. This confirms there’s no evidence that the evaluation depends on the specific reference model; the low bias and high agreement we report are genuine properties of the judges, rather than the artifacts of our particular reference set.
>
>
> [1] Yann Dubois, Xuechen Li, Rohan Taori, Tianyi Zhang, Ishaan Gulrajani, Jimmy Ba, Carlos Guestrin, Percy Liang, Tatsunori B. Hashimoto. 2023. AlpacaFarm: A Simulation Framework for Methods that Learn from Human Feedback. NeurIPS 2023.
>
> [2] Tianle Li, Wei-Lin Chiang, Evan Frick, Lisa Dunlap, Tianhao Wu, Banghua Zhu, Joseph E. Gonzalez, Ion Stoica. 2024. From Crowdsourced Data to High-Quality Benchmarks: Arena-Hard and BenchBuilder Pipeline. https://arxiv.org/abs/2406.11939
>
> [3] Lianmin Zheng, Wei-Lin Chiang, Ying Sheng, Siyuan Zhuang, Zhanghao Wu, Yonghao Zhuang, Zi Lin, Zhuohan Li, Dacheng Li, Eric P. Xing, Hao Zhang, Joseph E. Gonzalez, Ion Stoica. 2023. Judging LLM-as-a-Judge with MT-Bench and Chatbot Arena. NeurIPS 2023.
>
> > Final Comments
>
> We hope our response has addressed your initial concerns and we would appreciate if you could consider raising your appraisal in light of it.

---

> > ### Comment · Reviewer_2ui6 · 2025-11-25
> >
> > Thank you very much for the thorough reply and the detailed clarifications, that's very helpful.
> >
> > Just one comment regarding tool use: it might be worth to include a short discussion about it in the main paper (if space permits), since the community is getting more and more interested in this kind of evaluations.
> >
> > Happy to increase my score.

---

> > > ### Author Response · Authors · 2025-11-26
> > > **Followup reply to Reviewer 2ui6**
> > >
> > > Sure, we agree that the community is increasingly excited about tool use and will include a short discussion on it in the main paper.

---

### Author Response · Authors · 2025-12-02
**General Comment for Area Chair**

We thank the reviewers for their kind recognition of our contributions and constructive feedback. Out of the three reviewers (2ui6, knFu and pJFi) who have replied to our initial response, all three stated that our response has addressed their initial concerns. In addition, two reviewers who initially gave lower initial scores (2ui6 at 4 and pJFi at 6) have explicitly stated `Happy to increase my score.` and `I have raised my score accordingly` respectively, accompanied by a score increase from both reviewers, which were unfortunately reset due to the Nov 27 incident. These replies alongside their updated scores suggest that all four reviewers (2ui6 and pJFi as above; yfAJ and knFu who both gave initial score of 8) recommend acceptance for the paper. We hope the area chair can consider these perspectives in their final recommendations for the paper.

---

### Meta-Review · Area_Chair_QAYJ · 2026-01-07

**Summary:**

This paper introduces ProfBench, a benchmark of expert-authored response–criterion pairs across four professional domains, together with an LLM-judge framework aimed at enabling scalable and affordable evaluation of long-form professional reasoning. Reviewers agree that the benchmark addresses an important gap in LLM evaluation, moving beyond short-answer tasks toward document-grounded, real-world professional work.

The main concerns relate to domain scope, potential saturation and imbalance across domains, the reliability and interpretability of LLM-as-a-judge, and the inherent limitations of rubric-based evaluation. The rebuttal responds thoroughly, clarifying the rationale for domain selection, detailing the annotation and review process, adding robustness and fairness analyses for the LLM judges, and discussing future-proofing strategies. While some conceptual limitations remain, particularly the inability of rubric-based scoring to capture holistic qualities, these are clearly acknowledged and appropriately scoped.

Please address the following in the revised version: several figures/tables would benefit from careful editing to improve legibility and navigation.

Overall, I recommend weak accept. The benchmark is carefully constructed, empirically validated, and fills a meaningful gap in current evaluation practice. The remaining issues are largely about scope and presentation rather than fundamental flaws, and do not outweigh the paper’s contributions.

**Reviewer Concerns:**

I believe most reviewer concerns are largely addressed by the rebuttal, with a few remaining partially open.

Reviewer 2ui6: Scope, tool use, domain choice, and judge dependence are reasonably clarified. Robustness to reference models is supported by low bias and permutation tests. Long-term breadth and saturation remain future-work issues.

Reviewer yfAJ: Rubric creation and judging are much clearer, with added category-level analysis and qualitative failure patterns. Limitations of F1 are acknowledged, though deeper failure-mode analysis could help.

Reviewer knFu: Judge reliability, missing related work, and leakage concerns are addressed. However, the limitation of rubric-based evaluation vs. a holistic quality assessment remains partly resolved.

Reviewer pJFi: Adjudication transparency and fairness analysis are improved, with added diagnostics mitigating concerns about bias metrics and domain imbalance, though the presentation should remain concise.

**Reviewer Scores:**

With full discussion, I would expect modest positive score adjustments across reviewers, reflecting the thorough rebuttal, additional analyses, and improved clarity. While some higher-level limitations remain, they do not outweigh the benchmark’s value or the strength of the empirical validation. Overall, the paper meets the bar for acceptance.

---

### Decision · Program_Chairs · 2026-01-26

Accept (Poster)